# Encoding Unitig-level Assembly Graphs with Heterophilous Constraints for Metagenomic Contigs Binning

**Hansheng Xue**[1,2]**, Vijini Mallawaarachchi**[3]**, Lexing Xie**[1]**, Vaibhav Rajan**[2]*

[1]School of Computing, Australian National University, Canberra, Australia
[2]School of Computing, National University of Singapore, Singapore
[3]College of Science and Engineering, Flinders University, Adelaide, Australia
{hansheng.xue,lexing.xie}@anu.edu.au, vaibhav.rajan@nus.edu.sg
vijini.mallawaarachchi@flinders.edu.au

## Abstract

Metagenomics studies genomic material derived from mixed microbial communities in diverse environments, holding considerable significance for both human health and environmental sustainability. Metagenomic binning refers to the clustering of genomic subsequences obtained from high-throughput DNA sequencing into distinct bins, each representing a constituent organism within the community. Mainstream binning methods primarily rely on sequence features such as composition and abundance, making them unable to effectively handle sequences shorter than 1,000 bp and inherent noise within sequences. Several binning tools have emerged, aiming to enhance binning outcomes by using the assembly graph generated by assemblers, which encodes valuable overlapping information among genomic sequences. However, existing assembly graph-based binners mainly focus on simplified contig-level assembly graphs that are recreated from assembler's original graphs, *unitig-level assembly graphs*. The simplification reduces the resolution of the connectivity information in original graphs. In this paper, we design a novel binning tool named UNITIGBIN, which leverages representation learning on unitig-level assembly graphs while adhering to heterophilous constraints imposed by single-copy marker genes, ensuring that constrained contigs cannot be grouped together. Extensive experiments conducted on synthetic and real datasets demonstrate that UNITIGBIN significantly surpasses state-of-the-art binning tools.

## 1 Introduction

*Metagenomics* involves the analysis of genetic materials originating from mixed microbial communities present in various environments (Kaeberlein et al., 2002). It offers a suite of tools rooted in genome sequencing to address crucial questions related to human health and environmental sustainability. As an illustration, the Human Microbiome Project (Turnbaugh et al., 2007) uses metagenomic analysis techniques to acquire valuable insights into the intricate microbial communities residing in human body. This effort also aids in identifying microbial species associated with various diseases present in the human gut (Nayfach et al., 2019). In a standard metagenomic analysis workflow, genetic materials are gathered from the microbial community and then processed through a sequencing platform to generate DNA sequences commonly referred to as *reads*. Since these genetic materials are mixed together, it's uncertain which species each read belongs to. A core challenge in downstream analysis is to determine the species present within the input sample by examining these reads. However, these reads are too short for direct analysis. Many metagenomic techniques use *assembly graphs* to assemble these short reads into longer DNA sequences known as *unitigs*. *Contigs* are then formed by combining one or multiple connected unitigs (Xiang et al., 2023). In an assembly graph, each vertex corresponds to a unitig, and each edge represents the overlapping relationships between two unitigs (Nurk et al., 2017; Kolmogorov et al., 2020). Contigs are represented as either

---

*Corresponding author.

a single vertex or a path comprising several vertices within the assembly graph. *Contigs binning* refers to the clustering of these contigs into distinct bins corresponding to constituent genomes.

Many existing metagenomic binning tools rely on statistical information extracted from contigs themselves, including nucleotide composition and abundance features (Breitwieser et al., 2019; Yue et al., 2020). These tools do not take into account the homophily information within the assembly graph (Barnum et al., 2018), which suggests that sequences connected to each other in the assembly graph are more likely to belong to the same species. In addition, inherent noise within sequences presents additional challenges for these sequence feature-based binning methods. Several binning tools have been developed recently to leverage assembly graphs, including GraphBin (Mallawaarachchi et al., 2020) and GraphMB (Lamurias et al., 2022). However, instead of directly using unitig-level assembly graphs produced by the assembler, they simplify original graphs and re-construct contig-level assembly graphs, where vertices represent contigs and edges represent their overlaps. However, the simplification reduces the resolution of the connectivity information in unitig-level assembly graphs and may introduce erroneous edges (Xiang et al., 2023). Moreover, many existing binning tools face difficulties and exhibit low recall values when handling short sequences, often shorter than 1,000 bp, which are commonly excluded from analysis.

Additional biological information, such as single-copy marker genes, can also be exploited to improve binning results (Albertsen et al., 2013; Dupont et al., 2012). Single-copy marker genes are genes that occur only once in each species. If two contigs share the same single-copy marker gene, it is highly likely that they belong to different species. Some binning tools have utilized this additional information from single-copy marker genes to estimate the initial number of bins or enhance the quality of contig binning results (Mallawaarachchi & Lin, 2022; Lamurias et al., 2023). However, there are limited graph neural network models that can be directly employed to model the unitig-level assembly graph with heterophilous relationships. Moreover, dealing with the large-scale characteristics of unitig-level assembly graphs in real metagenomic data poses significant challenges for the learning process. In this paper, we develop a graph neural network framework designed to model the unitig-level assembly graph while also adhering to heterophilous constraints imposed by single-copy marker genes, called UNITIGBIN. The contributions of this paper are listed as follows:

- To the best of our knowledge, this is the first use of graph neural networks to model the unitig-level assembly graph in the field of metagenomic contig binning.
- We devise a novel model for constraint-based graph learning, UNITIGBIN-*Learning*, which captures the unitig-level assembly graph with constraints by employing a diffusive convolution and optimizing triplet constraints. A $p$-batch strategy is designed for parallelization.
- We devise a novel UNITIGBIN-*Binning* framework that leverages a *Matching* algorithm to initialize markered contigs, uses *Propagating* labels to annotate unmarked contigs meanwhile satisfying constraints, and employs a local *refining* strategy to fine-tune final binning.
- Extensive experiments on synthetic and real datasets show that UNITIGBIN significantly outperforms state-of-the-art binners in terms of binning purity and completeness, regardless of whether graphs are from standard metagenomic assemblers, metaSPAdes and metaFlye.

## 2 RELATED WORK

**Contigs binners.** Despite contigs being assembled from short reads using assembly graphs, the majority of existing binners overlook the homophily information present in these assembly graphs. Instead, these binning tools rely on composition (normalized oligonucleotide, *i.e.*, short strings of length $k$, frequencies) and coverage (average number of reads aligning to each position of the contig) information to perform contig binning. For example, MetaWatt (Strous et al., 2012) leverages multivariate statistics and Markov models to bin contigs. CONCOCT (Alneberg & et al., 2014) combines the variational Bayesian model selection and Gaussian mixture model to cluster contigs into bins. MaxBin2 (Wu et al., 2016) designs an expectation-maximization algorithm that uses both composition and coverage information to iteratively bin the contigs. BusyBeeWeb (Laczny et al., 2017) is a web application that uses a bootstrapped supervised binning approach for contig binning. MetaBAT2 (Kang et al., 2019) is a graph partitioning approach that uses contigs' composition to construct the graph. SolidBin (Wang et al., 2019) uses a semi-supervised spectral clustering algorithm combined with additional biological knowledge. MetaBinner (Wang et al., 2023) is an ensemble binning tool that can integrate various types of features. In addition, several binning tools have been

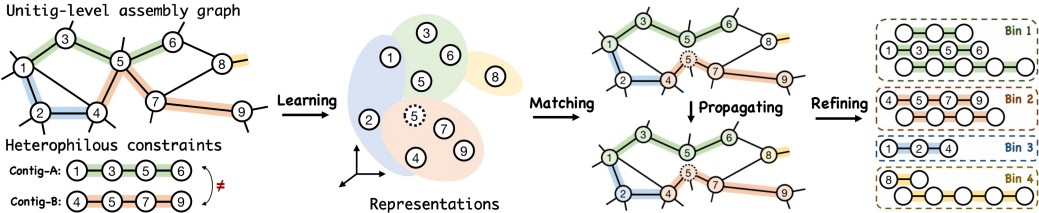

Figure 1: The framework of UNITIGBIN consists of *Learning*, *Matching*, *Propagating*, and *Refining*.

developed to enhance performance using deep learning techniques. For instance, VAMB (Nissen et al., 2021) uses deep variational autoencoders to learn both composition and coverage information. CLMB (Zhang et al., 2022) employs deep contrastive learning techniques to produce robust results even from noisy data. SemiBin (Pan et al., 2022) designs a semi-supervised Siamese neural network incorporating must-link and cannot-link constraints obtained from reference genomes. These binning tools do not utilize assembly graphs and often omit short contigs because composition and coverage features are less reliable for short contigs, leading to lower recall values.

**Assembly graph improves binning.** To enhance the binning outcomes, recent bin-refinement methods (Mallawaarachchi et al., 2020) have introduced the utilization of assembly graphs. However, these bin-refinement tools are not independent and require the initial binning results from existing binners as a starting point. MetaCoAG (Mallawaarachchi & Lin, 2022) is a standalone binning tool capable of integrating composition, abundance, and assembly graph information to enhance binning performance. In addition, several methods have been developed to use graph neural networks to model the assembly graph. For instance, GraphMB (Lamurias et al., 2022) uses a variational autoencoder model to encode both composition and abundance and then feeds these features into graph neural networks for contigs binning. This approach does not incorporate additional information like heterophilous constraints from single-copy marker genes. RepBin (Xue et al., 2022) designs a self-supervised graph learning framework for modeling assembly graphs while encoding prior constraints. Then, a semi-supervised label propagation model is employed for contig binning. CCVAE (Lamurias et al., 2023) develops a variational autoencoder to simultaneously learn the assembly graph and the information of single-copy marker genes and then uses a clustering algorithm for contig binning. A common limitation of these graph-based binning tools is their applicability solely to contig-level assembly graphs, which are generated from the unitig-level assembly graph using specific strategies. The transition from unitig-level to contig-level reduces the resolution of connectivity information in the original graph and may introduce errors due to the chosen strategy.

## 3 METHODOLOGY

**Preliminaries.** Given a unitig-level assembly graph $\mathcal{G}=(V, E, P, X)$ along with its constraints $\mathcal{C}$. The output embedding for unitigs and contigs in the graph are $d$-dimensional vectors $Z \in \mathbb{R}^{|V| \times d}$ and $\hat{Z} \in \mathbb{R}^{|P| \times d}$, and contigs binning results are denoted as $\mathcal{B}=\{b_i \in \mathbb{R}^K, i \in |P|\}$, where $K$ denotes the number of bins. In the assembly graph $\mathcal{G}$, $V$ is the nodes or unitigs set, $E=\{(v_i, v_j)|v_i, v_j \in V\}$ denotes edges indicating the overlap between unitigs, $P=\{(v_i, ...v_j, ...v_k)|v_i, v_j, v_k \in V\}$ corresponds to the paths or contigs within the graph, and $X=\{x_v|v \in V\}$ are features associated with nodes. Heterophilous constraints from the single-copy marker genes are $\mathcal{C}=\{(p_i, ..., p_j)|p_i, p_j \in P\}$. The objective of binning metagenomic contigs is to assign a label $b_i \in \mathcal{B}$ to each contig $p_i \in P$.

**Preprocessing.** The heterophilous constraints $\mathcal{C}$ are created by employing the FragGeneScan (Rho et al., 2010) and HMMER (Eddy, 2011) tools to detect contigs containing single-copy marker genes, following a similar approach as MaxBin (Wu et al., 2016) and MetaCoAG (Mallawaarachchi & Lin, 2022). When contigs belong to the same constraint set, it implies that these contigs should not be grouped together in pairs within the bins. The unitig-level assembly graphs are constructed from two widely used assemblers: metaSPAdes (Nurk et al., 2017) and metaFlye (Kolmogorov et al., 2020).

Prior to modeling the assembly graph and clustering contigs, we initially perform preprocessing operations based on known knowledge (unitig-level assembly graph $\mathcal{G}$ and heterophilous constraints $\mathcal{C}$). Two operators are introduced in UNITIGBIN, i) *Graph disentangling* and ii) *Contigs sampling*. *Graph disentangling* is designed to separate contigs within the unitig-level assembly graph. For example, when constraints suggest that contig $A$ and contig $B$ should belong to distinct bins, yet these

two contigs share an overlapping unitig in the assembly graph (as shown in Figure 1), we create a new unitig by duplicating the original unitig 5 to disentangle the assembly graph. *Contigs sampling* is devised to create positive relationships among contigs by leveraging the inherent structure of the assembly graph. Different from conversion strategies used in GraphBin and GraphMB to construct contig-level assembly graphs. Here, we focus on determining whether two paths are directly connected or linked by only a single hop (Miller et al., 2010); in this case, we establish a positive edge between these contigs. The sampled positive contigs set is represented as $\mathcal{O} = \{(p_i, p_j)|p_i, p_j \in P\}$.

**Overview.** UNITIGBIN consists of two main components: *Learning*, which uses a graph neural network to model the unitig-level assembly graph while adhering to constraints, and *Binning*, a contig-level framework. In the *Binning* stage, a *Matching* algorithm is employed to initialize markered contigs, *Propagating* labels are used to annotate unmarked contigs while satisfying constraints, and a local *Refining* strategy is incorporated to fine-tune binning assignments (refer to Figure 1).

### 3.1 *Learning*: REPRESENTING UNITIG-LEVEL ASSEMBLY GRAPH WITH CONSTRAINTS

The UNITIGBIN-*Learning* model aims to obtain latent representations for both unitigs/nodes $Z$ and contigs/paths $\hat{Z}$ considering both unitig-level assembly graph $\mathcal{G}$ and heterophilous constraints $\mathcal{C}$. In this section, we will introduce the *Learning* framework in three components, a) Diffusion encoder-decoder framework, b) Triplet Gaussian constraints optimization, and c) $p$-Batch parallelization.

#### 3.1.1 DIFFUSION ENCODER-DECODER FRAMEWORK

An encoder-decoder architecture is adopted as the foundational learning framework in *Learning*, which comprises two primary components: a graph diffusive convolution encoder (Klicpera et al., 2019) and an inner-product decoder. The diffusive encoder captures the graph's topology and initial node features, while the inner-product decoder reconstructs the graph's structure using learned features from the diffusive encoder. Minimizing the reconstruction loss, which measures the

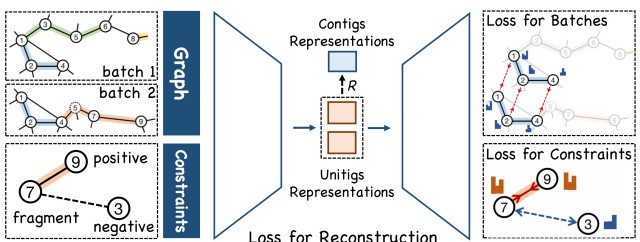

Figure 2: The framework of the UNITIGBIN-*Learning*.

dissimilarity between the original and reconstructed graph, allows us to obtain the node embeddings.

**Encoder-Decoder.** In *Learning*, a variational autoencoder (Kingma & Welling, 2013) is established. $A$ denotes the adjacency matrix of a unitig-level assembly graph with self-loops ($A{=}A{+}I_N$, $I_N$ is the unit matrix), $D$ stands for its diagonal degree matrix, *i.e.*, $D_{ii}{=}\sum_{j=1}^{N} A_{ij}$. We use DIFFCONV to symbolize the diffusive convolution. Then, the diffusive encoder can be formulated as: $q(Z|X, A){=}\prod_{i=1}^{N} q(z_i|X, A)$, with $q(z_i|X, A){=}\mathcal{N}(z_i|\mu_i, \text{diag}(\sigma_i^2))$, where $\mu{=}\text{DIFFCONV}_\mu(H, A)$, $\log \sigma{=}\text{DIFFCONV}_\sigma(H, A)$, and $H{=}\text{DIFFCONV}(X, A)$. The inner-product decoder is calculated as $p(\hat{A}|Z){=}\prod_{i=1}^{N}\prod_{j=1}^{N} p(A_{ij}|z_i, z_j)$, with $p(A_{ij}{=}1|z_i, z_j){=}\text{Sigmoid}(z_i^\top z_j)$. The object is as follows:

$$\mathcal{L}_g = \mathbb{E}_{q(Z|X,A)}[\log p(\hat{A}|Z)] - KL[q(Z|X,A)||p(Z)], \tag{1}$$

where $p(Z){=}\prod_i p(z_i){=}\prod_i \mathcal{N}(z_i|0, I)$ is a Gaussian prior. A weighted cross entropy loss (Kipf & Welling, 2016) is used in Equation 1 to measure the reconstruction error between $A$ and $\hat{A}$. $KL(\cdot)$ is the divergence function to measure the similarity between the distribution of $q(Z|X, A)$ and $p(Z)$.

**Diffusive convolution.** Following RepBin (Xue et al., 2022), we also use the PageRank-based diffusion (Page et al., 1999) to model the unitig-level assembly graph. To provide a brief explanation, when considering a node $i$ with vectorial feature $x_i$, the iterative calculation of its diffusive feature follows the equation $\text{PPR}(x_i){=}(1{-}\alpha)A\text{PPR}(x_i){+}\alpha x_i$, where $\alpha \in (0, 1]$ is the probability of transitioning to a different state. The transitions state for node $i$ can be computed as $\text{PPR}(x_i){=}\alpha(I_N{-}(1{-}\alpha)A)^{-1}x_i$. The diffusive convolution and layer-wise propagation rule can be formulated as:

$$H^{l+1} = \sigma(\text{DIFFCONV} \cdot H^l\Theta^l), \quad \text{DIFFCONV} = \alpha[I_N - (1-\alpha)D^{-1/2}AD^{-1/2}]^{-1} \tag{2}$$

where $\sigma(\cdot)$ denotes a non-linear activation function, and $\Theta^l \in \mathbb{R}^{|V| \times d_l}$ is a $l$-layer trainable transformation matrix, $d_l$ is the embeddings dimension in the $l$-layer, and $H^0{=}X$. By optimizing the object

in Equation 1, latent embeddings for unitigs can be obtained, $Z \in \mathbb{R}^{|V| \times d}$. A readout function can be used to generate features for contigs, $\hat{Z} = \mathcal{R}(Z)$, with $\hat{Z}_i = \frac{1}{|P_i|} \sum_{v \in P_i} Z_v$, and $\hat{Z} \in \mathbb{R}^{|P| \times d}$.

### 3.1.2 TRIPLET GAUSSIAN CONSTRAINTS OPTIMIZATION

In constraints $\mathcal{C}$, each set signifies that certain contigs contain the identical marker gene and these contigs must not be grouped pairwise into the same bin. In *Learning*, we convert contig-constraints $\mathcal{C}$ into the pairwise contig-constraints set $\mathcal{C}'$ and pairwise unitig-constraints set $\mathcal{M}'$. In each pairwise constraint $(i, j) \in \mathcal{M}'$, it indicates that unitig $i$ and $j$ must not be assigned to the same bin. We treat these pairwise constraints as negative samples, whereas we sample existing edges in the graph as positive samples. In detail, for every pairwise constraint $(i, j) \in \mathcal{M}'$, we sample node $i$'s neighbors, $N_i$, as positive edges. Sampled triplet constraints can be defined as $\mathcal{M} = \{(i, j, k), i, j \in V, k \in N_i\}$. To integrate Gaussian distributions and triplet constraints, we draw inspiration from (Bojchevski & Günnemann, 2018) and incorporate *measuring* and *ranking* strategies. Within the encoder-decoder framework, the Gaussian embeddings in hidden layers can be acquired as follows: $z_i = \mathcal{N}(\mu_i, \sum_i)$ with $\sum_i = diag(\text{elu}(\log \sigma_i) + 1)$, $\mu_i \in \mathbb{R}^d$, $\sum_i \in \mathbb{R}^{d \times d}$, where $\mu = \text{DIFFCONV}_\mu(H, A)$ and $\log \sigma = \text{DIFFCONV}_\sigma(H, A)$. The KL divergence-based dissimilarity measurement (He et al., 2015) between two Gaussian embeddings $z_i$ and $z_j$ can be represented as $\Delta(z_i, z_j) = D_{KL}(\mathcal{N}_j || \mathcal{N}_i) = \frac{1}{2} [tr(\sum_i^{-1} \sum_j) + (\mu_i - \mu_j)^\top \sum_i^{-1} (\mu_i - \mu_j) - d - \log \frac{|\sum_j|}{|\sum_i|}]$, where $tr(\cdot)$ and $|\cdot|$ denote the trace and determinant of a matrix respectively. Each triplet constraint $(i, j, k) \in \mathcal{M}$ signifies that unitig $i$ and $j$ must not be in the same bins, and there is a high probability that nodes $i$ and $k$ should belong to the same bin. In other words, node $i$ is more closely related to $k$ compared to node $j$. We formulate the triplet constraints ranking strategy as $\Delta(z_i, z_k) < \Delta(z_i, z_j)$. The square-exponential loss (LeCun et al., 2006) is used to measure the triplet constraints ranking as:

$$\mathcal{L}_c = \sum_{(i,j,k) \in \mathcal{M}} [D_{KL}(\mathcal{N}_k || \mathcal{N}_i)^2 + \exp^{-D_{KL}(\mathcal{N}_j || \mathcal{N}_i)}] \tag{3}$$

---

**Algorithm 1:** The Unitig-level Assembly Graph Learning Algorithm UNITIGBIN-*Learning*.

---

**Data:** Unitig-level assembly graph $\mathcal{G}$; constraints $\mathcal{C}$; dimension of embedding $d$; number of graph batches $n$;

**Result:** Embedding for unitigs $Z$ and contigs $\hat{Z}$.

1   $\mathcal{G}, \mathcal{O} \leftarrow \text{Preprocess}(\mathcal{G}, \mathcal{C})$   `// Graph untangling and Contigs sampling`
2   $\mathcal{M} \leftarrow \text{Sample}(\mathcal{G}, \mathcal{C})$   `// Sample triplet unitig constraints`
3   $\text{Batches} \leftarrow p\text{-Batch}(\mathcal{O}, n)$   `// Split batches`
4   **for** $e \in \text{epochs}$ **do**
5     **for** $b \in \text{Batches}$ **do**
6       $H_b \leftarrow \text{DIFFCONV}(A_b, X_b)$   `// Base diffusive convolution`
7       $\mu_b, \log \sigma_b \leftarrow \text{DIFFCONV}_\mu(H_b, A_b), \text{DIFFCONV}_\sigma(H_b, A_b)$ `//Gaussian embedding`
8       $\mathcal{L}_{g_b} \leftarrow \text{Equation 1}$   `// Compute loss for graph reconstruction`
9     **end**
10    $\mathcal{L}_g \leftarrow \mathcal{L}_{g_b}$   `// Accumulate the batch losses`
11    $\mathcal{L}_c \leftarrow \text{Equation 3}, \mathcal{L}_b \leftarrow D_{KL}$   `// compute the constraint, batch loss`
12    $\mathcal{L} \leftarrow \mathcal{L}_g + \mathcal{L}_b + \lambda_1 \cdot \mathcal{L}_c$   `// Compute loss in Equation 4`
13   **end**
14   $Z \leftarrow \mu$   `// Unitigs Embedding`
15   $\hat{Z} \leftarrow \mathcal{R}(Z)$   `// Contigs Embedding`

---

### 3.1.3 $p$-BATCH: TRAINING DATA BATCHING

Training GNNs on unitig-level assembly graphs from real metagenomic data, which can reach millions in size, presents significant computational challenges. Creating training batches presents a challenge as it must satisfy two criteria: i) processing each contig in parallel while preserving its completeness, and ii) grouping diverse contigs with positive relationships into the same batch to retain this valuable information. To address these hurdles, we introduce a graph splitting and training module named $p$-Batch, which systematically selects independent sets of nodes from the *Positive-contig Graph* which derived from the positive contigs set $\mathcal{O}$ in an iterative manner. The $p$-Batch module takes each path as the minimum splitting unit and functions iteratively through two steps: i)

selecting the largest contigs sets from the candidates, and ii) feeding them into the smallest batch. The $p$-Batch will continue until all candidate contigs are fed into one batch. In practice, there are still some unitigs that are fed into different batches. We design a loss function to minimize the probability distribution of these jointly nodes. The objective function for the $p$-Batch loss function can be calculated as $\mathcal{L}_b = \sum_{(i,j) \in Q} D_{KL}(\mathcal{N}_j || \mathcal{N}_i)^2$, where $Q$ is the set of joint-unitigs pairs.

**Objective function.** The object of *Learning* is determined by a combination of $\mathcal{L}_g$, $\mathcal{L}_c$, and $\mathcal{L}_b$, with $\lambda_1$ regulating the significance of constraints loss. Refer to Algorithm 1 for pseudocode of *Learning*.

$$\mathcal{L} = \mathcal{L}_g + \mathcal{L}_b + \lambda_1 \cdot \mathcal{L}_c \tag{4}$$

### 3.2 *Binning*: COMPRISING MATCHING CONSTRAINTS, PROPAGATING AND REFINING BINS

***Matching*.** After obtaining the embeddings of contigs in the *Learning* step, you can directly apply existing clustering algorithms (such as K-Means) for contigs binning. However, dealing with imbalanced bin sizes adds complexity to the contigs binning process. RepBin (Xue et al., 2022) proposes a semi-supervised label propagation model using constrained contigs as initial labels. However, RepBin runs K-Means on embeddings of a large number of constrained contigs to initialize labels, which can be computationally expensive. The lack of a known number of bins is another challenge.

In UNITIGBIN, we devised a simple yet efficient matching algorithm for attaining optimal binning initialization. *Matching* mainly consists of two key steps: i) Binning Initialization and ii) Iterative Matching. Initially, we arrange constraints in $\mathcal{C}$ in descending order based on their length, selecting the largest set as the initial bin. We then perform iterative calculations to determine the similarity between matched bins and candidate contigs. A greedy method is used to select the maximum value for matching operations. In the matching process, we incorporate a threshold value denoted as $\mathcal{T}$. If the similarity between a bin and a contig is above $\mathcal{T}$, we add this contig to the bin. Instead, we opt to create a new bin that includes this contig. Refer to Algorithm 2 in Appendix A.1 for the pseudocode.

***Propagating*.** From the preceding *Learning* and *Matching* phases, we obtain embeddings of unitigs denoted as $Z \in \mathbb{R}^{|V| \times d}$ and initial labels assigned to constrained contigs represented by $Y_{\mathcal{C}} \in \mathbb{R}^{\bar{K}}$. We follow RepBin and design a contig-level label propagation model instead of running K-Means algorithm directly. Besides, we also introduce a penalty function to maximize constraint satisfaction.

*Propagating* consists of three parts: graph convolution, readout function, and fully connected layer. Graph convolution learns both the unitig-level assembly graph and unitigs features from *Learning*, which can be described as $Z^{l+1} = \sigma(\text{CONV} \cdot Z^l \Theta^l)$, where $\text{CONV} = D^{-1/2} A D^{-1/2}$. The embeddings of contigs can be obtained through the readout function, $\hat{Z} = \mathcal{R}(Z)$, $\hat{Z} \in \mathbb{R}^{|P| \times d}$. Then, the binning probability can be represented as $Y = \text{Softmax}(\hat{Z}W + b)$, $Y \in \mathbb{R}^{|P| \times K}$, where $K$ is the number of bins. A cross-entropy function is used to optimize the binning results. However, the binning assignment may violate prior constraints. To maximize constraint satisfaction, we introduce an optimization function. Given $K$ bins, we use a $0/1$ matrix in $\mathbb{R}^{K \times K}$ for incorporating constraints. The constraint matrix $\mathcal{I}_{\neq}$ denotes the binary conflict relationships among $K$ bins, i.e., $\mathcal{I}_{\neq}(i,j) = 1$ if $i \neq j$ and 0 otherwise, for any $i, j \in \{1, ..., K\}$. The bin-assignment matrix $Y \in \mathbb{R}^{|P| \times K}$ is a matrix that represents the bin assignment probability (over $K$ bins) for each contig $i$ in its corresponding row $Y_i$. For any constraint $(i,j) \in \mathcal{C}'$, we aim to assign different bins to $i$ and $j$ and thus maximize the sum of joint-probabilities with different bins, i.e., $Y_i^{\mathsf{T}} \mathcal{I}_{\neq} Y_j$. The objective function is as follows:

$$\mathcal{L} = -\sum_{l \in Y_{\mathcal{C}}} \sum_{k=1}^{K} Y_{lk} \ln Z_{lk} - \lambda_2 \cdot \frac{1}{|\mathcal{C}'|} \sum_{(i,j) \in \mathcal{C}'} \log(Y_i^{\mathsf{T}} \mathcal{I}_{\neq} Y_j) \tag{5}$$

***Refining*.** In *Refining*, our primary goal is to explore potential binning assignments for contigs, taking into account heterophilous constraints. This step primarily consists of two components: i) *Splitting* and ii) *Merging*. *Splitting* aims to divide existing bins into multiple sub-bins when identical marker genes are present within the bin. *Merging* is intended to combine sub-bins into a larger bin when these sub-bins do not share the same marker genes. Refer to Appendix A.1 for the pseudocode.

## 4 EXPERIMENTS

**Datasets and Baselines.** We evaluate UNITIGBIN model on 12 datasets, consisting of 6 assembled by metaSPAdes v3.15.2 (Nurk et al., 2017) and 6 assembled by metaFlye v2.9 (Kolmogorov et al.,

Table 1: CheckM results for the number of HQ bins by UNITIGBIN and baselines.

| Methods | Sim20G | Sim50G | Sim100G | Sharon | DeepHPM | COPD |
|---|---|---|---|---|---|---|
| MetaBAT2 | 5 | 16 | 3 | 2 | 0 | 0 |
| MaxBin2 | **20** | 35 | 54 | 6 | 8 | 9 |
| Semi Bin | 18 | 38 | 68 | 5 | - | - |
| VAMB | 18 | 31 | 51 | 5 | 2 | 6 |
| GraphMB | 9 | 13 | 18 | 2 | - | - |
| CCVAE | 12 | 15 | 28 | 2 | - | - |
| RepBin | 18 | 15 | 19 | 1 | 0 | - |
| MetaCOAG | 17 | 34 | 69 | **7** | 8 | 17 |
| UNITIGBIN | **20** | **43** | **76** | **7** | **12** | **21** |
| △% MaxBin2 | *0%* | *18.6%* | *28.9%* | *14.3%* | *33.3%* | *57.1%* |
| △% SemiBin/VAMB | *10%* | *11.6%* | *10.5%* | *28.6%* | *83.3%* | *71.4%* |
| △% MetaCoAG | *15%* | *20.9%* | *9.2%* | *0%* | *33.3%* | *19.0%* |

2020). In metaSPAdes-assembled datasets, Sim20G, Sim50G, and Sim100G are three datasets collected from GraphBin2 (Mallawaarachchi et al., 2021) and MetaCoAG (Mallawaarachchi & Lin, 2022). In metaFlye-assembled datasets, 6 real-world Wastewater Treatment Plant (WWTP) datasets are collected (Singleton et al., 2021). Table A1 provides a comprehensive overview of the dataset statistics. UNITIGBIN is evaluated against three categories of binning tools: a) 2 traditional approaches, MaxBin 2.0 (Wu et al., 2016) and MetaBAT2 (Kang et al., 2019); b) 2 deep learning-based binning tools, SemiBin (Pan et al., 2022) and VAMB (Nissen et al., 2021); c) 4 assembly graph-based binning models, GraphMB (Lamurias et al., 2022), RepBin (Xue et al., 2022), Meta-CoAG (Mallawaarachchi & Lin, 2022), and CCVAE (Lamurias et al., 2023).

**Metrics and Experimental Settings.** We use the popular CheckM v1.1.3 (Parks et al., 2015) tool to evaluate the binning results of UNITIGBIN and baselines. CheckM assesses bin quality through sets of single-copy marker genes and without using ground truth. We use CheckM to assess the completeness and contamination of the bins generated by each tool. For metaSPAdes datasets, we adhere to the experimental setup outlined in MetaCoAG (Mallawaarachchi & Lin, 2022). We define precision as $1/(1 + \mathrm{contamination})$ and recall as completeness. High-quality (HQ) bins are characterized by precision $> 90$ and recall $> 80$. Medium-quality (MQ) bins have precision $> 80$ and recall $> 50$, while the remaining bins are classified as Low-quality (LQ) bins. For metaFlye datasets, we follow the experimental setup used in GraphMB (Lamurias et al., 2022) and CCVAE (Lamurias et al., 2023), which employs two specific criteria: completeness $> 90$ & contamination $< 5$, and completeness $> 50$ & contamination $< 10$, to assess the quality of bins. We also use AMBER v2.0.2 (Meyer et al., 2018) tool and calculate the Precision, Recall, F1, Adjusted Rand Index (ARI) metrics (Xue et al., 2022) to evaluate simulated datasets using ground truth.

## 4.1 EVALUATION ON METASPADES-BASED DATASETS

Table 1 shows that UNITIGBIN significantly outperforms state-of-the-art baselines, achieving the highest number of high-quality bins as evaluated by CheckM. In Sim100G, UNITIGBIN yields 76 high-quality bins, approximately 9.2% more than the highest number obtained by baselines (69 for MetaCoAG). In COPD, UNITIGBIN also attains the highest number of high-quality (HQ) bins, with 21 HQ bins, which is considerably greater than the second-highest number of HQ bins obtained by MetaCoAG (17). This substantial gap between UNITIGBIN and baselines underscores the superior performance of our model in contig binning. The CheckM results for medium-quality (MQ) bins generated by UNITIGBIN and baselines can be found in Table A3. UNITIGBIN consistently outperforms other methods by achieving the highest number of HQ+MQ bins across most datasets.

In three simulated datasets, we also employ the AMBER tool and calculate the Precision, Recall, F1, and ARI scores to assess the performance of both UNITIGBIN and baselines. Here we take Sim20G as an example, Figure 3 denotes the Average Completeness (AC) and Average Purity (AP) at the nucleotide level, while Table 2 presents the F1 and ARI score (with 'bp' representing the nucleotide-level and 'seq' representing the sequence-level), and the number of HQ bins. Comparison with baselines demonstrates that UNITIGBIN achieves the highest level of performance. In particular, UNITIGBIN is capable of binning not only long contigs but also those shorter than

1,000 bp, which are typically discarded by other binning tools. For instance, UNITIGBIN achieves a sequence-level F1 score of 0.952, which is significantly higher than the second-highest F1 score obtained by MaxBin2, 0.632. The calculated Precision, Recall, F1, and ARI scores are shown in Table A2.

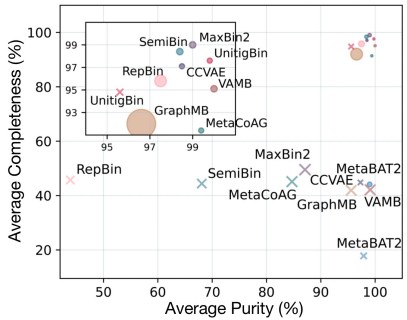

Figure 3: AC and AP for Sim20G.

Table 2: F1(%), ARI(%), and HQ metrics on the Sim20G.

| Methods | F1(bp) | F1(seq) | ARI(bp) | ARI(seq) | HQ↑ |
|---------|--------|---------|---------|----------|-----|
| MetaBAT2 | 61.0 | 30.1 | 39.2 | 21.6 | 4 |
| MaxBin2 | **99.0** | 63.2 | 99.0 | 77.5 | **20** |
| SemiBin | 98.4 | 53.7 | 98.1 | 41.7 | 19 |
| VAMB | 97.5 | 59.1 | 97.9 | 96.7 | 18 |
| GraphMB | 94.2 | 58.3 | 55.9 | 34.4 | 10 |
| CCVAE | 97.8 | 61.3 | 79.1 | 46.8 | 13 |
| RepBin | 96.6 | 44.8 | 96.3 | 14.2 | 16 |
| MetaCoAG | 95.3 | 58.9 | 99.1 | 78.7 | 15 |
| UNITIGBIN | 98.7 | **95.2** | **99.3** | **97.4** | **20** |

## 4.2 EVALUATION ON METAFLYE-BASED DATASETS

We also benchmark UNITIGBIN and baselines on six real datasets assembled using metaFlye. We use the CheckM tool and count the number of bins that meet two criteria (following CCVAE): *A)* completeness>90 & Contamination<5; and *B)* completeness>50 & Contamination<10. Figure 4 clearly shows that UNITIGBIN outperforms baselines across all six datasets. UNITIGBIN produces a total of 1,775 bins that meet the criteria *A*. In contrast, the highest count achieved by baselines is 962 bins, which is 45.8% less than the number of bins obtained by UNITIGBIN. Notably, CCVAE uses CheckM to detect single-copy marker genes within contigs and extract heterophilous constraints. Moreover, CCVAE also employs CheckM to evaluate binning results. To eliminate potential ambiguity, we follow the pipeline of MaxBin2 and MetaCoAG, using FragGeneScan and HMMER to identify contigs containing marker genes. We also present results for UNITIGBIN using constraints extracted from CheckM, which are detailed in Figure A1. In summary, UNITIGBIN consistently demonstrates superior performance across datasets assembled by both metaSPAdes and metaFlye.

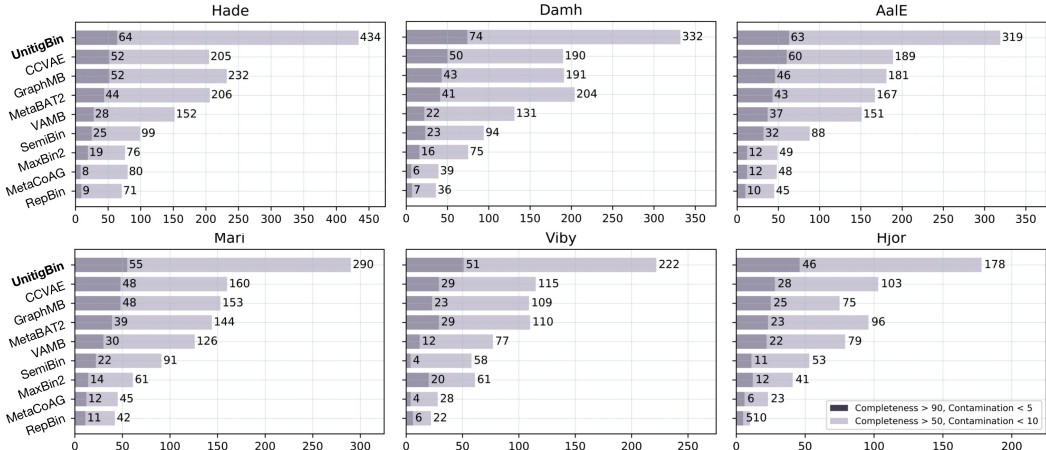

Figure 4: CheckM results of UNITIGBIN and baselines on 6 real datasets assembled using metaFlye.

## 4.3 VISUALIZATION AND EXPERIMENTAL ANALYSIS

**Visualization.** To gain deeper insights into the binning results, we employ the python-igraph package to visualize the unitig-level assembly graph of Sim100G, alongside the ground truth and binning results obtained from various binning tools (selected ten representative bins, see Figure 5). Nodes represent unitigs, while edges indicate overlapping relationships between distinct unitigs. Distinct colors represent different species or bins. UNITIGBIN produces binning results that align well with the ground truth, whereas other baselines struggle with missing or inaccurate labels.

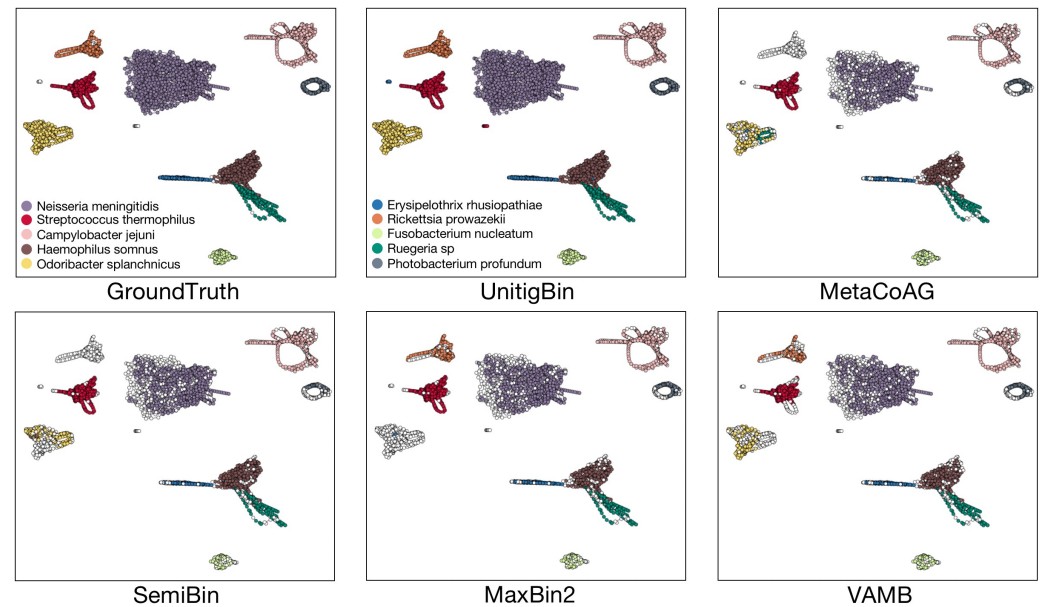

Figure 5: Visualization of selected ten bins in Sim100G with ground truth and different binners.

**Ablation study** & **Parameters analysis**. To assess the effectiveness of our proposed model, we perform an ablation study to investigate the individual algorithmic components within UNITIGBIN. Figure 6 illustrates that each component within UNITIGBIN contributes to the improvement in contigs binning performance (more details in Appendix A.7). We also analyze the impact of parameters such as dimension $d$, the transition probability $\alpha$ in diffusive convolution, the threshold $\mathcal{T}$ in *Matching*, the importance of constraints $\lambda_1$ in loss function Eqn 4, and the importance of constraints $\lambda_2$ in loss function Eqn 5. Figures A3 show that UNITIGBIN displays a relatively low sensitivity to variations in the aforementioned parameters. As $\lambda_1$ is raised, involving more importance of constraints, the performance of UNITIGBIN increases and tends to stabilize.

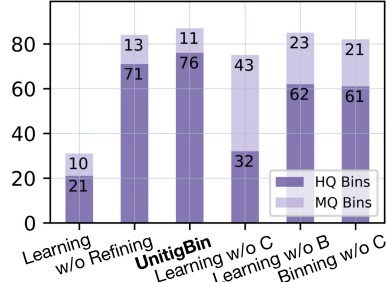

Figure 6: Results of UNITIGBIN and its variants on Sim100G.

**Training process** & **Running time**. Figure A2 (a) and (b) show the training process of *Learning* and *Propagating* in UNITIGBIN respectively. As the number of training iterations increases, the proportion of violated constraints decreases and more constraints are satisfied. We also benchmark the running time of UNITIGBIN against selected baselines on Sim100G (refer to Figure A2 (c)). UNITIGBIN is the second-fastest deep learning-based binning tool, with a runtime of approximately 30 mins, beaten only by VAMB, which is faster. It is significantly faster than other deep learning-based methods.

## 5 CONCLUSION

To model the unitig-level assembly graph directly output from metagenomic assemblers while incorporating heterophilous constraints derived from single-copy marker genes, we present a novel binning tool called UNITIGBIN, a graph neural network model with constraint satisfaction designed for binning metagenomic contigs. UNITIGBIN comprises *Learning*, which uses a graph neural network model to learn the unitig-level assembly graph while adhering to constraints. It is followed by a contig *Binning* framework that employs an adapted *Matching* algorithm to initialize markered contigs, uses *Propagating* to annotate unmarked contigs while satisfying constraints, and incorporates a local *Refining* strategy to fine-tune binning assignments. Extensive experiments conducted on both synthetic and real datasets show that UNITIGBIN outperforms existing binning tools significantly. The primary limitations of the model include its inability to label overlapping bins, where contigs belong to multiple species, and the difficulty of binning unmarked and short contigs. As future work, we plan to explore graph neural networks for binning short, unmarked contigs with multiple labels, and efficiently encoding large-scale unitig-level assembly graphs.

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

# A APPENDIX

## A.1 THE PSEUDOCODE FOR UNITIGBIN

In this paper, we propose the model of neural networks for constraint-based graph representation to solve the metagenomic contigs binning problem, called UNITIGBIN. UNITIGBIN consists of two main components: *Learning*, which uses a graph neural network model to directly learn the unitig-level assembly graph while adhering to constraints, and *Binning*, a contig-level binning framework. In the *Binning* stage, a *Matching* algorithm is employed to initialize markered contigs, *Propagating* labels are used to annotate unmarked contigs while satisfying constraints, and a local *Refining* strategy is incorporated to fine-tune binning assignments. In the subsequent subsection, we will elucidate the *Matching* and *Refining* modules and provide the pseudocode.

*Matching*. In *Matching*, we design an adapted matching algorithm to obtain optimal binning initialization. The *Matching* process primarily has two steps: i) Binning Initialization and ii) Iterative Matching. Initially, we arrange the constraints in $\mathcal{C}$ in descending order based on their length, selecting the largest constraint set as the initial bin. Then, we perform iterative calculations to determine the similarity between matched bins and candidate contigs. We employ a greedy approach to select the maximum value for matching operations. During the matching process, we incorporate a threshold value denoted as $\mathcal{T}$. If the similarity between a bin and a candidate is below this threshold, we consider it unfavorable to add the candidate contig to the bin. Instead, we opt to create a new bin that includes this contig. The pseudocode for the UNITIGBIN-*Matching* is as follows.

---

**Algorithm 2:** The Markers Matching Algorithm UNITIGBIN-*Matching*.

---

**Data:** Embedding for contigs $\hat{Z}$; Heterophilous constraints $\mathcal{C}$; Threshold for adjustment $\mathcal{T}$;
**Result:** Matched binning initializations $\mathrm{Y}_{\mathcal{C}}$.

1   $\mathcal{C} \leftarrow \mathrm{Sort}(\mathcal{C})$   `// Sort `$\mathcal{C}$` in descending order by their length`
2   $\mathrm{Y}_{\mathcal{C}} \leftarrow \mathcal{C}[0]$   `// Initialize bin with maximum set of constraints`
3   **for** $k \in [1, |\mathcal{C}|)$ **do**
4      $S = \{(i,j) : s\} \leftarrow \mathrm{Sim}(\mathrm{Y}_{\mathcal{C}}, \mathcal{C}[k])$ `// Compute sim between bins and cands`
5      $S = \{(i,j) : s\} \leftarrow \mathrm{Sort}(S)$ `// Sort `$S$` in descending order by sim value`
6      $\mathrm{Tag} = []$   `// Initialize the iteration tag as None`
7      **for** $\{(i,j) : s\} \in S$ **do**
8          **if** $i, j$ *not in Tag* **then**
9              **if** $s < \mathcal{T}$ **then**
10                 $\mathrm{Y}_{\mathcal{C}}[|\mathrm{Y}_{\mathcal{C}}|] \leftarrow j$   `// Add a new bin with contig j`
11                 $\mathrm{Tag} \leftarrow |\mathrm{Y}_{\mathcal{C}}|, j$   `// Add bin `$|\mathrm{Y}_{\mathcal{C}}|$` and contig j to iteration tag list`
12              **else**
13                 $\mathrm{Y}_{\mathcal{C}}[\mathrm{i}] \leftarrow \mathrm{j}$   `// Assign contig j to bin i`
14                 $\mathrm{Tag} \leftarrow i, j$   `// Add bin i and contig j to iteration tag list`
15              **end**
16          **end**
17      **end**
18 **end**

---

*Refining*. In *Refining*, our primary goal is to explore potential binning assignments for contigs, taking into account heterophilous constraints. This step primarily consists of two components: i) *Splitting* and ii) *Merging*. *Splitting* aims to divide existing bins into multiple sub-bins when identical marker genes are present within the bin. *Merging* is intended to combine sub-bins into a larger bin when these sub-bins do not share the same marker genes. The pseudocode for the UNITIGBIN-*Refining* is as follows.

## A.2 DATASETS AND BASELINES

**Datasets.** We evaluate UNITIGBIN model on 12 datasets, consisting of 6 assembled by metaSPAdes v3.15.2 (Nurk et al., 2017) and 6 assembled by metaFlye v2.9 (Kolmogorov et al., 2020). Within the metaSPAdes-assembled datasets, Sim20G, Sim50G, and Sim100G are three datasets collected

from GraphBin2 (Mallawaarachchi et al., 2021) and MetaCoAG (Mallawaarachchi & Lin, 2022)[1], which are simulated based on the species found in the simMC+ dataset (Wu et al., 2014). Paired-end reads for these datasets are simulated using InSilicoSeq (Gourlé et al., 2019). Sharon, COPD, and DeepHPM are 3 real-world datasets. Sharon is a preborn infant gut metagenome (Sharon et al., 2013) identified with the National Center for Biotechnology Information (NCBI) accession number *SRA052203*. COPD refers to the metagenomics of the Chronic Obstructive Pulmonary Disease Lung Microbiome (Cameron et al., 2016) and is associated with the NCBI BioProject number *PRJEB9034*. DeepHPM represents a human metagenome sample from the tongue dorsum of a participant in the Deep WGS HMP clinical samples (Lloyd-Price et al., 2017), with the NCBI accession number *SRX378791*. In the metaFlye-assembled datasets, 6 real-world datasets are collected from GraphMB (Lamurias et al., 2022) and CCVAE (Lamurias et al., 2023)[2]. The six datasets consist of real-world Wastewater Treatment Plant (WWTP) datasets obtained from (Singleton et al., 2021) under the BioProject number *PRJNA629478*. FragGeneScan (Rho et al., 2010) and HMMER (Eddy, 2011) are applied to detect contigs containing marker genes and generate heterophilic constraints, following the methodology outlined in MetaCoAG (Mallawaarachchi & Lin, 2022) and MaxBin 2.0 (Wu et al., 2016). Ground truths for three simulated datasets (Sim20/50/100G) are derived using Minimap2 (Li, 2018) to map the contigs to the reference genomes, still following the steps of MetaCoAG. Table A1 provides comprehensive statistics of 12 datasets.

---

**Algorithm 3:** The Binning Refinement Algorithm UNITIGBIN-*Refining*.

**Data:** Binning for contigs $\mathcal{B}$; Heterophilous constraints $\mathcal{C}$;

**Result:** Refined binning $\hat{\mathcal{B}}$.

```
1  // Splitting
2  C2M ← C     // Construct a mapping function:Contig→Marker
3  B̄ ← []      // Initialize the splitting binset
4  for bin ∈ B do
5      bs,ms ← [],[]     // Initialize the subbin and submarker
6      for c ∈ bin do
7          m ← C2M(c)      // Map the contig to corresponding marker
8          if m ∉ ms then
9              ms←ms+m, bs←bs+b      // No identical marker
10         else
11             B̄ ← B̄+bs      // Identical marker, add a new subbin
12             bs←b, ms←m      // Initialize the subbin and submarker
13         end
14     end
15     B̄ ← B̄+bs      // add a bin
16 end
17 // Merging
18 B2M ← B̄, C   // Construct a mapping function:Bin→Marker
19 Tag ← True   // Initialize the iteration tag as True
20 while Tag == True do
21     Tag ← False   // Set the iteration tag as False
22     S={(i,j):s} ← Count(B2M(B̄)) // Count common markers between
        pairwise bins
23     for {(i,j):s} in S do
24         if s == 0 then
25             B̄ ← Merging(B̄,i,j)   // Merge bin i and bin j into one bin
26             Tag ← True   // Set the iteration tag to be true
27             break
28         end
29     end
30 end
31 B̂ ← B̄   // Output the final binning assignment
```

---

[1] https://figshare.com/projects/MetaCoAG/121014
[2] https://zenodo.org/records/6122610

Table A1: Summary and statistics of the 12 datasets.

| Dataset | No. of Samples | Fragments length (bp) | No. of Fragments | Contigs length (bp) | No. of Contigs | No. of Links | No. of Markers | Assembler |
|---|---|---|---|---|---|---|---|---|
| Sim-20G | 1 | 10,891.04 | 6,131 | 46,234.75 | 1,452 | 8,213 | 75 | |
| Sim-50G | 1 | 5,888.21 | 24,401 | 28,680.50 | 5,058 | 32,654 | 85 | |
| Sim-100G | 1 | 4,501.06 | 67,516 | 19,977.80 | 15,729 | 90,124 | 85 | metaSPAdes |
| Sharon | 18 | 1,039.83 | 37,164 | 1,212.44 | 29,899 | 29,174 | 109 | |
| DeepHPM | 8 | 594.81 | 227,635 | 694.53 | 130,363 | 126,497 | 108 | |
| COPD | 18 | 716.74 | 452,600 | 759.00 | 125,524 | 109,134 | 108 | |
| Hjor | 4 | 39,443.96 | 22,005 | 42,677.75 | 21,111 | 5,335 | 108 | |
| Viby | 4 | 48,103.74 | 27,719 | 52,183.60 | 26,427 | 7,342 | 109 | |
| Damh | 4 | 48,748.20 | 40,048 | 54,275.15 | 37,285 | 12,977 | 110 | metaFlye |
| Mari | 4 | 40,005.90 | 42,539 | 44,377.07 | 40,450 | 12,178 | 108 | |
| AalE | 4 | 41,192.55 | 46,938 | 44,788.51 | 44,777 | 11,749 | 109 | |
| Hade | 4 | 36,665.14 | 82,063 | 41,467.02 | 76,924 | 26,737 | 104 | |

**Baselines.** UNITIGBIN is evaluated against three categories of binning tools: a) 2 traditional approaches, MaxBin 2.0 (Wu et al., 2016) and MetaBAT2 (Kang et al., 2019); b) 2 deep learning-based binning tools, SemiBin (Pan et al., 2022) and VAMB (Nissen et al., 2021); c) 4 assembly graph-based binning models, GraphMB (Lamurias et al., 2022), RepBin (Xue et al., 2022), Meta-CoAG (Mallawaarachchi & Lin, 2022), and CCVAE (Lamurias et al., 2023). For all baselines, we fine-tune their models with various parameters and present metric scores of their best configurations.

**i) Traditional approaches**:
*MaxBin2*: is a probabilistic model that incorporates an Expectation-Maximization algorithm, utilizing both composition and coverage features for contigs binning (Wu et al., 2016). Available at `https://sourceforge.net/projects/maxbin2/`.
*MetaBAT2*: constructs a graph using the composition information of contigs and employs a graph partitioning algorithm to address the metagenomic binning problem (Kang et al., 2019). Available at `https://bitbucket.org/berkeleylab/metabat/src/master/`.

**ii) Deep learning-based binning tools**:
*SemiBin*: is a reference-based binning approach that aligns contigs to reference genomes, creating must-link and cannot-link constraints. It then utilizes a semi-supervised siamese neural network to integrate prior constraints into the contig binning process (Pan et al., 2022). Available at `https://github.com/BigDataBiology/SemiBin`.
*VAMB*: uses deep variational autoencoder models to encode both the composition and coverage information of contigs and then employs an iterative clustering algorithm to bin contigs (Nissen et al., 2021). Available at `https://github.com/RasmussenLab/vamb`.

**iii) Assembly graph-based binning models**:
*GraphMB*: encodes contig composition and abundance information using a variational autoencoder model, combining these learned features with the assembly graph to train a graph neural network model. The resulting contig embeddings are then fed into the same clustering algorithm as VAMB for contig binning (Lamurias et al., 2022). Available at `https://github.com/MicrobialDarkMatter/GraphMB`.
*RepBin*: employs a constraint-based self-supervised graph learning framework to model the assembly graph, followed by the utilization of a GCN-based label propagation model for contig binning (Xue et al., 2022). Available at `https://github.com/xuehansheng/RepBin`.
*MetaCoAG*: designs an algorithm that integrates composition, abundance, and assembly graph to enhance contigs binning. Available at `https://github.com/metagentools/MetaCoAG`.
*CCVAE*: incorporates the prior constraints of single-copy marker genes, identified using CheckM, into a variational autoencoder model to learn the assembly graph. It subsequently employs a clustering algorithm on learned embeddings for contig binning (Lamurias et al., 2023). Available at `https://github.com/MicrobialDarkMatter/ccvae`.

A.3 IMPLEMENTATION DETAILS

In this section, we will provide instructions on how to run our proposed UNITIGBIN model to reproduce the experimental results presented in our paper. In the preprocessing phase, two crucial operations are carried out to create the unitig-level assembly graphs and derive heterophilous constraints. These constraints are generated by identifying single-copy marker genes within contigs.

**a) Create Unitig-level Assembly Graphs.** Two different types of assembly graphs are employed in this paper. We utilize two widely used assemblers, namely metaSPAdes (Nurk et al., 2017) and metaFlye (Kolmogorov et al., 2020), to generate the unitig-level assembly graphs. The example command is as follows:
i) The metaSPAdes assembler:
```
$ spades --meta -1 Reads_1.fastq -2 Reads_2.fastq -o /output_dir
```
ii) The metaFlye assembler:
```
$ flye --meta --pacbio-raw Reads.fastq -o /output_dir
```

**b) Extract Heterophilous Constraints.** In UNITIGBIN, we use FragGeneScan (Rho et al., 2010) and HMMER (Eddy, 2011) to extract the constraints by identifying single-copy marker genes in contigs. The example command is as follows:
```
$ run_FragGeneScan.pl -genome=/path/to/contigs.fasta
-out=/output_dir/contigs.fasta.frag -complete=0
-train=complete -thread=8 1>/output_dir/contigs.fasta.frag.out
2>/output_dir/contigs.fasta.frag.err
$ hmmsearch --domtblout /output_dir/contigs.fasta.hmmout --cut_tc
--cpu 8 /path/to/marker.hmm /output_dir/contigs.fasta.frag.faa
1>/output_dir/contigs.fasta.hmmout.out
2>/output_dir/contigs.fasta.hmmout.err
```
Next, we employ the code from SolidBin (Wang et al., 2019) to extract constraints and store them in several sets. Each set represents the contigs containing a specific single-copy marker gene.

**c) Run UNITIGBIN.** To reproduce the experimental results in this paper, we can simply run the following command (we take the Sim20G as an example):
```
$ python unitigbin.py --data Sim20G --epoch 2000 --dim 32 --lr
0.01 --nbatch 1 --α 0.05 --λ₁ 0.7 --λ₂ 0.1 --th 0.01
```
where parameter `--data` represents the selected dataset; `--epoch` denotes the number of epoch; `--dim` is the dimension of the embedding; `--lr` is the learning rate; `--nbatch` represents the number of batches; `--`$\alpha$ denotes the probability of transition in diffusive convolution; `--th` represents the threshold in the *Matching*; `--`$\lambda_1$ and `--`$\lambda_2$ are two parameters control the importance of constraints in the process of *Learning* and *Propagating* respectively.

**d) CheckM Evaluation.** After obtaining the binning results, we can use CheckM v1.1.3 (Parks et al., 2015) to evaluate the performance of binning. The example command is as follows:
```
$ checkm lineage_wf -x fasta /input_dir/*.fasta
/output_dir/CheckM_Res
$ checkm analyze -x fasta /path/to/checkm_data_2015_01_16/hmms/
phylo.hmm /output_dir /output_dir/CheckM_Res
$ checkm qa --out_format 1 -f /output_dir/CheckM_Res/result.txt
/path/to/checkm_data_2015_01_16/hmms/phylo.hmm /output_dir/CheckM_Res
```

**Running environment.** UNITIGBIN is implemented in Python 3.6 and Pytorch 1.8 using the Linux server with Intel Xeon Platinum 8268 2.9 GHz CPU, 96GB RAM and 1 Nvidia Tesla Volta V100-SXM2 with 32GB memory.

**Experimental setups.** For UNITIGBIN, we use composition features from the sequences and adjacency matrix of the unitig-level assembly graph as the initial features for nodes. The default settings of UNITIGBIN hyperparameters are as follows. The representation dimensions are all empirically set and vary from 32 to 256. The parameter $\alpha$ in the diffusive convolution is simply set within the range of [0.005, 0.05]. The batch size of the $p$-batch varies from 1 to 5 in all datasets. Following the experimental setup in (Mallawaarachchi & Lin, 2022) and (Lamurias et al., 2023), all the algorithms run five times on each input dataset and the best binning assignment is reported. The UNITIGBIN model is freely available at `https://github.com/xuehansheng/UnitigBIN`.

Table A2: Evaluation results by UNITIGBIN and baselines.

| Dataset | Methods | Contigs / Ratio↑ | Bins | Precision↑ | Recall↑ | F1↑ | ARI↑ |
|---|---|---|---|---|---|---|---|
| Sim20G (K=20) | MetaBAT2 | 402 / 28.0% | 88 | 97.01 | 7.96 | 14.70 | 21.55 |
| | MaxBin2 | 666 / 46.3% | 21 | 89.48 | 42.97 | 58.06 | 77.46 |
| | VAMB | 530 / 36.9% | 21 | **99.62** | 38.29 | 55.32 | 96.52 |
| | SemiBin | 790 / 54.9% | 20 | 68.21 | 38.29 | 49.05 | 41.69 |
| | GraphMB | 732 / 50.9% | 34 | 74.35 | 39.33 | 51.45 | 34.43 |
| | CCVAE | 737 / 51.3% | 51 | 85.25 | 38.59 | 53.13 | 46.82 |
| | RepBin | 1,437 / 99.9% | 20 | 97.99 | 95.84 | 96.90 | 94.51 |
| | MetaCoAG | 595 / 41.4% | 20 | 89.52 | 38.74 | 54.07 | 78.66 |
| | UNITIGBIN | **1,438 / 100%** | 21 | 98.22 | **98.07** | **98.14** | **97.38** |
| Sim50G (K=50) | MetaBAT2 | 1,139 / 22.5% | 222 | 81.13 | 6.09 | 11.33 | 14.98 |
| | MaxBin2 | 1,900 / 37.6% | 49 | 84.23 | 33.59 | 48.03 | 76.32 |
| | VAMB | 1,310 / 25.9% | 46 | 88.74 | 31.35 | 46.33 | 85.74 |
| | SemiBin | 2,231 / 44.1% | 241 | **90.67** | 36.63 | 52.18 | 74.18 |
| | GraphMB | 2,332 / 46.1% | 87 | 58.24 | 38.35 | 46.25 | 18.11 |
| | CCVAE | 2,137 / 42.2% | 65 | 54.83 | 36.94 | 44.14 | 16.78 |
| | RepBin | **5,058 / 100%** | 47 | 78.69 | 90.80 | 84.31 | 77.55 |
| | MetaCoAG | 2,030 / 40.1% | 47 | 79.72 | 35.32 | 48.95 | 69.30 |
| | UNITIGBIN | **5,058 / 100%** | 48 | 89.41 | **91.06** | **90.23** | **90.61** |
| Sim100G (K=100) | MetaBAT2 | 54 / 0.4% | 32 | **100.00** | 0.46 | 0.92 | **83.95** |
| | VAMB | 3,814 / 24.9% | 87 | 87.01 | 29.82 | 44.42 | 81.13 |
| | MaxBin2 | 5,723 / 37.4% | 95 | 71.10 | 25.84 | 37.90 | 52.43 |
| | GraphMB | 8,045 / 52.5% | 226 | 46.11 | 39.62 | 42.62 | 7.65 |
| | CCVAE | 7,206 / 47.0% | 243 | 56.94 | 36.58 | 44.54 | 13.35 |
| | SemiBin | 6,706 / 43.8% | 957 | 91.40 | 34.46 | 50.05 | 71.77 |
| | RepBin | 15,310 / 99.9% | 72 | 63.51 | **92.55** | 75.33 | 48.58 |
| | MetaCoAG | 6,033 / 39.4% | 90 | 74.84 | 30.31 | 43.15 | 59.89 |
| | UNITIGBIN | **15,319 / 100%** | 101 | 76.77 | 78.45 | **77.60** | 79.85 |

## A.4 EVALUATION METRICS

In addition to the CheckM (Parks et al., 2015) and AMBER (Meyer et al., 2018) tools, we also calculate the Precision, Recall, F1, and ARI as evaluation metrics for the contigs binning task. $A \in \mathcal{R}^{K \times S}$ is used to denote the confusion matrix, where $K$ is the number of bins predicted by binners and $S$ represents the number of true species in the ground truth. The element in $A$, like $A_{k,s}$, indicates that the number of contigs are clustered into the bin $k$ but actually belong to the species $s$. $N$ is the total number of contigs binned by this tool and $N_u$ denotes the number of contigs that are not clustered or discarded by this binning tool. The detailed evaluation metrics are described as follows:

$$Precision = \frac{\sum_k \max_s A_{k,s}}{\sum_k \sum_s A_{k,s}}, \; Recall = \frac{\sum_s \max_k A_{k,s}}{\sum_k \sum_s A_{k,s} + N_u}, \tag{6}$$

$$F1 = 2 \times \frac{Precision \times Recall}{Precision + Recall}, \; ARI = \frac{\sum_{k,s} \binom{A_{k,s}}{2} - \frac{\sum_k \binom{A_{k,\cdot}}{2} \sum_s \binom{A_{\cdot,s}}{2}}{\binom{N}{2}}}{\frac{1}{2}(\sum_k \binom{A_{k,\cdot}}{2} + \sum_s \binom{A_{\cdot,s}}{2}) - \frac{\sum_k \binom{A_{k,\cdot}}{2} \sum_s \binom{A_{\cdot,s}}{2}}{\binom{N}{2}}}. \tag{7}$$

In the results, we present the number of contigs identified by different binning tools and also calculate the proportion of contigs that have been labeled. The higher the proportion of labeled contigs, the better the binning tool performs. Besides, we also show the number of bins identified by different binners. Both predicted bins higher or lower than ground truth are not good. From the evaluation metrics collected in Table A2, we can find that UNITIGBIN achieves the highest score among the most evaluation metrics. In particular, UNITIGBIN can label both long and short contigs, leading to a significantly higher Recall compared to other binning tools, with the exception of RepBin. For

example, in the Sim100G dataset, MetaBAT2 can only label 0.4% of the contigs, while UNITIGBIN is capable of labeling all 15,319 contigs.

## A.5   CHECKM RESULTS ON METASPADES DATASETS.

We use CheckM tool to evaluate the performance of UNITIGBIN on 6 metaSPAdes-assembled datasets. We adhere to the experimental setup outlined in MetaCoAG (Mallawaarachchi & Lin, 2022). We define precision as $1/(1 + \text{contamination})$ and recall as completeness. High-quality (HQ) bins are characterized by precision > 90 and recall > 80. Medium-quality (MQ) bins have precision > 80 and recall > 50, while the remaining bins are classified as Low-quality (LQ) bins. The detailed evaluation metrics are collected in Table A3. Several deep learning-based binning tools, such as SemiBin, GraphMB, CCVAE, and RepBin, could not be executed on the DeepHPM and COPD datasets due to resource limitations. Table A3 demonstrates that UNITIGBIN outperforms the other baseline methods significantly. UNITIGBIN achieves the highest number of HQ bins and HQ+MQ bins. For instance, in the real COPD dataset, UNITIGBIN obtains 21 HQ and 28 MQ bins, which is more than the second-highest number of HQ bins achieved by MetaCoAG (17 for HQ and 25 MQ bins). In DeepHPM, UNITIGBIN achieves 12 HQ bins, surpassing the highest number of HQ bins obtained by baselines (8 for MetaCoAG and MaxBin2).

Table A3: Evaluation results by UNITIGBIN and baselines (CheckM).

| Methods | Sim20G | | | Sim50G | | | Sim100G | | |
|---|---|---|---|---|---|---|---|---|---|
| | HQ↑ | MQ↑ | LQ↓ | HQ↑ | MQ↑ | LQ↓ | HQ↑ | MQ↑ | LQ↓ |
| MetaBAT2 | 5 | **11** | 72 | 16 | **12** | 194 | 3 | 5 | 24 |
| MaxBin2 | **20** | 0 | 1 | 35 | 3 | 7 | 54 | **15** | 26 |
| SemiBin | 18 | 1 | 3 | 38 | 4 | 8 | 68 | 9 | 31 |
| VAMB | 18 | 1 | 3 | 31 | 3 | 12 | 51 | 10 | 26 |
| GraphMB | 9 | 1 | 24 | 13 | 1 | 73 | 18 | 2 | 206 |
| CCVAE | 12 | 1 | 38 | 15 | 0 | 71 | 28 | 2 | 213 |
| RepBin | 18 | 1 | 1 | 15 | 1 | 15 | 19 | 8 | 33 |
| MetaCoAG | 17 | 0 | **0** | 34 | 6 | 3 | 69 | 8 | **13** |
| UNITIGBIN | **20** | 0 | 1 | **43** | 4 | 4 | **76** | 11 | 14 |

| | Sharon | | | DeepHPM | | | COPD | | |
|---|---|---|---|---|---|---|---|---|---|
| | HQ↑ | MQ↑ | LQ↓ | HQ↑ | MQ↑ | LQ↓ | HQ↑ | MQ↑ | LQ↓ |
| MetaBAT2 | 2 | 2 | 20 | 0 | 1 | 60 | 0 | 2 | 74 |
| MaxBin2 | 6 | 2 | 6 | 8 | 14 | 47 | 9 | 24 | 123 |
| SemiBin | 5 | 0 | 10 | - | - | - | - | - | - |
| VAMB | 5 | 1 | 4 | 2 | 3 | 13 | 6 | 7 | 48 |
| GraphMB | 2 | 0 | 53 | - | - | - | - | - | - |
| CCVAE | 2 | 0 | 74 | - | - | - | - | - | - |
| RepBin | 1 | 0 | 8 | 0 | 4 | 46 | - | - | - |
| MetaCoAG | **7** | 3 | **0** | 8 | 10 | **9** | 17 | 25 | 26 |
| UNITIGBIN | **7** | **5** | 2 | **12** | **17** | 10 | **21** | **28** | **20** |

## A.6   CONSTRAINTS FROM CHECKM

In the UNITIGBIN model, we use the FragGeneScan (Rho et al., 2010) and HMMER (Eddy, 2011) tools to generate heterophilous constraints, following the MaxBin2 (Wu et al., 2016) and Meta-CoAG (Mallawaarachchi & Lin, 2022). However, it's worth noting that CCVAE (Lamurias et al., 2023) relies on the CheckM tool to generate prior information for constraints. To further assess the performance of our proposed UNITIGBIN model, we also integrate the heterophilous constraints obtained from CheckM instead of the original constraints into UNITIGBIN. Figure A1 shows the number of HQ (completeness>90 & Contamination<5) and MQ (completeness>50 & Contamination<10) bins achieved by UNITIGBIN under the constraints of CheckM. UNITIGBIN(CheckM) achieves more HQ and MQ bins than UNITIGBIN and significantly more than CCVAE. For example, in the Hade dataset, UNITIGBIN(CheckM) achieves the highest number of HQ bins (75), surpassing the HQ bins obtained by CCVAE (52 HQ bins) and UNITIGBIN (64 HQ bins).

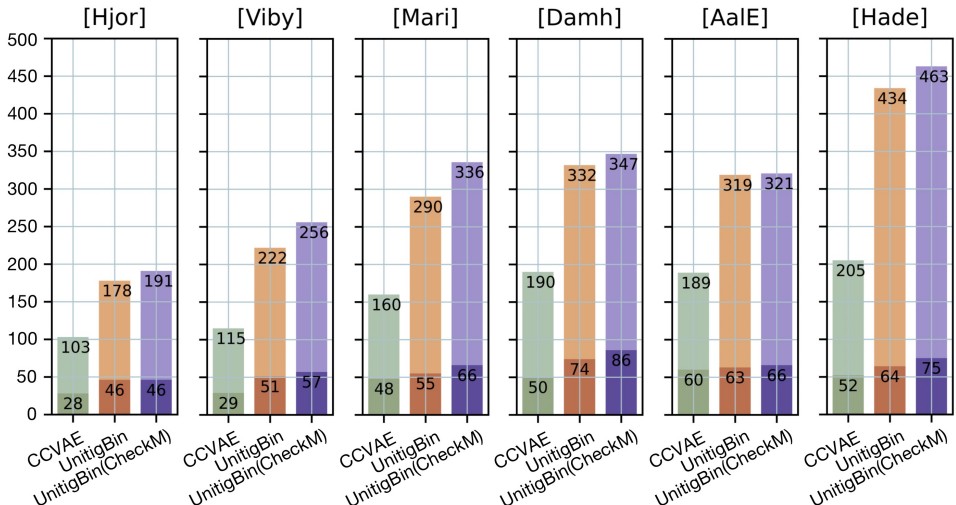

Figure A1: Results of CCVAE, UNITIGBIN and UNITIGBIN(CheckM) on 6 datasets.

### A.7 PARAMETERS ANALYSIS AND SCALABILITY

**More details in ablation study.** In Figure 6, the '*Learning*' denotes the *Learning* model and uses K-Means to bin contigs; '*w/o Refining*' represents the UNITIGBIN model without the final *Refining* step; '*Learning w/o C*' indicates that not incorporate constraints into the *Learning* process; '*Learning w/o B*' means the *Learning* model without batch loss; and '*Binning w/o C*' is the *Binning* model does not optimize constraints satisfaction. Figure 6 illustrates that each component within UNITIG-BIN contributes to the improvement in contig binning performance.

**Training and running time.** Figure A2 (a) and (b) show the training process of *Learning* and *Propagating* modules in UNITIGBIN respectively. The loss value stabilizes when the number of epochs reaches 600. Besides, as the number of training iterations increases, the proportion of violated constraints decreases. We also benchmark the running time of UNITIGBIN against selected baselines on Sim100G (refer to Figure A2 (c)). MaxBin2 and MetaCoAG are two non-deep learning models and were executed on CPU, while other deep learning-based binning tools were trained on GPU. Here, we provide the running time and the count of HQ bins. UNITIGBIN is the second-fastest deep learning-based binning tool, with a runtime of approximately 30 mins, beaten only by VAMB, which is faster. It is significantly faster than other deep learning-based binning tools, including GraphMB (49 mins), SemiBin (55 minutes), RepBin (55 minutes), and CCVAE (72 minutes).

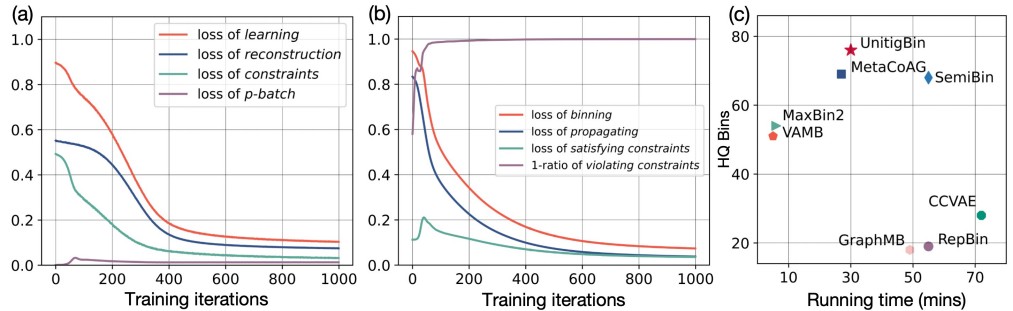

Figure A2: The training loss and running time of UNITIGBIN on Sim100G.

**Parameters analysis.** In this part, we investigate the importance of core parameters in the UNITIG-BIN without *Refining* model, including the dimension of embedding $d$, the probability of transition in diffusive convolution $\alpha$, the threshold in the matching algorithm $\mathcal{T}$, and two parameters control the importance of constraints $\lambda_1$ and $\lambda_2$. Figure A3 (a) shows that UNITIGBIN is relatively robust to the dimension of embedding $d$ ranging from 32 to 512. In Figure A3 (b), we observe a grad-

ual increase in the number of HQ bins as the $\alpha$ parameter in DIFFCONV is varied from $5e^{-4}$ to $5e^{-2}$. Figure A3 (c) shows that UNITIGBIN is also relatively robust to the parameter of threshold in the matching algorithm. The parameter $\lambda_1$ in the *Learning* model governs the significance of heterophilous constraints. As $\lambda_1$ increases, the number of HQ bins also increases, reaching its maximum when set to 0.7. The UNITIGBIN model demonstrates robustness to changes in the parameter $\lambda_2$ in the *Binning* model (as shown in Figure A3 (e)).

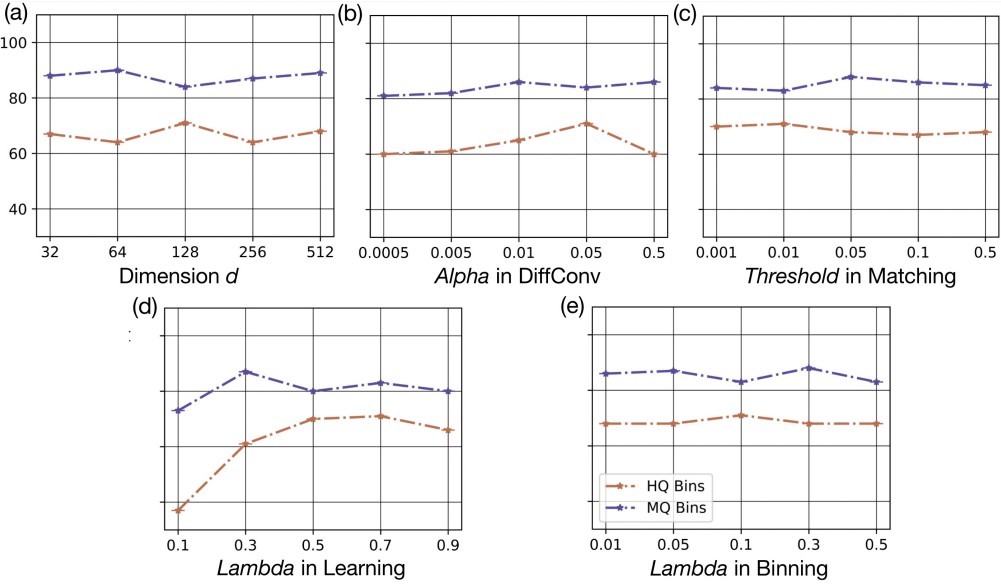

Figure A3: Parameters analysis of UNITIGBIN on Sim100G ($d$, $\alpha$, $\mathcal{T}$, $\lambda_1$, and $\lambda_2$).

## A.8 ADDITIONAL EXPERIMENTAL RESULTS

**Evaluating constraints satisfaction.** One goal of our proposed UNITIGBIN model is to enhance the satisfaction of heterophilous constraints derived from single-copy marker genes. In order to assess the achievement of our objectives, we calculate the ratio of final binning results that deviate from heterophilous constraints. The illustration in Figure A3(b) depicts the procedure for minimizing the number of violating constraints. Furthermore, the table below (on Sim100G) provides the ratio of violating constraints accomplished by the UNITIGBIN and selected baseline methods. UNITIGBIN satisfies almost all constraints, exhibiting only a 0.9% violation of heterophilous constraints, a rate lower than that observed in other baseline methods.

Table A4: The specific proportion of violated heterophilous constraints.

|  | MaxBin2 | SemiBin | VAMB | MetaBAT2 | GraphMB | CCVAE | RepBin | MetaCoAG | UNITIGBIN |
|---|---|---|---|---|---|---|---|---|---|
| Num. | 1,712 / 127,449 | 2,226 / 180,044 | 2,623 / 152,948 | 37 / 1,313 | 42,244 / 170,476 | 22,627 / 167,211 | 8,854 / 177,910 | 2,069/ 153,026 | 1,686 / 182,366 |
| Ratio | 1.34% | 1.24% | 1.92% | 2.82% | 24.80% | 13.53% | 4.98% | 1.35% | 0.90% |

**Additional comparisons with the Metadecoder tool.** Metadecoder (Liu et al., 2022) is a binning tool based on a Gaussian mixture model, utilizing k-mer frequencies and coverages. Here, we also introduce Metadecoder as a baseline and conduct initial comparisons on three simulated datasets and one Sharon dataset. We followed the experimental settings specified in MetaCoAG and selected CheckM as the evaluation tool. As observed in Table A5, Metadecoder's performance is noticeably inferior to that of UNITIGBIN. In Sim20G, Metadecoder generates 19 high-quality bins, surpassing the count achieved by MetaCoAG (17) but falling short of UNITIGBIN (20). In the Sim50G and Sim100G datasets, Metadecoder produces 25 high-quality bins for Sim50G and 35 for Sim100G, respectively. These counts are notably lower than those achieved by UNITIGBIN (43 for Sim50G

and 76 for Sim100G). In the Sharon dataset, Metadecoder generates only 4 high-quality bins, a count significantly lower than the number achieved by both MetaCoAG and UNITIGBIN (7).

Table A5: Evaluation results by UNITIGBIN and baselines.

| Methods | Sim20G | | | Sim50G | | | Sim100G | | | Sharon | | |
|---|---|---|---|---|---|---|---|---|---|---|---|---|
| | HQ↑ | MQ↑ | LQ↓ | HQ↑ | MQ↑ | LQ↓ | HQ↑ | MQ↑ | LQ↓ | HQ↑ | MQ↑ | LQ↓ |
| Metadecoder | 19 | 0 | 2 | 25 | **10** | 16 | 35 | **27** | 45 | 4 | 1 | 11 |
| MetaCoAG | 17 | 0 | **0** | 34 | 6 | 3 | 69 | 8 | **13** | 7 | 3 | **0** |
| UNITIGBIN | **20** | 0 | 1 | **43** | 4 | 4 | **76** | 11 | 14 | 7 | 5 | 2 |

**Additional comparisons with the SimBin2 tool.** we conduct a comparison between UNITIGBIN and SemiBin2 (Pan et al., 2023), an enhanced version of SemiBin, to further assess performance. SemiBin2 is applied to three simulated datasets and one real world dataset, and its performance is compared with SemiBin, MetaCoAG (the best assembly graph-based baseline), and UNITIGBIN. We adhere to the experimental setup outlined in MetaCoAG (refer to A.5), and the resulting experimental data is presented in Table A6. SemiBin2 demonstrates superior performance compared to SemiBin and MetaCoAG across all three simulated datasets, although it falls slightly short of UNITIGBIN. In the Sharon dataset, SemiBin2, MetaCoAG, and UNITIGBIN all attain the maximum count of high-quality bins (7), which is higher than the number obtained by SemiBin (5).

Table A6: Evaluation results by UNITIGBIN and baselines.

| Methods | Sim20G | | | Sim50G | | | Sim100G | | | Sharon | | |
|---|---|---|---|---|---|---|---|---|---|---|---|---|
| | HQ↑ | MQ↑ | LQ↓ | HQ↑ | MQ↑ | LQ↓ | HQ↑ | MQ↑ | LQ↓ | HQ↑ | MQ↑ | LQ↓ |
| SemiBin | 18 | 1 | 3 | 38 | 4 | 8 | 68 | 9 | 31 | 5 | 0 | 10 |
| SemiBin2 | 19 | **2** | 1 | 41 | 3 | 6 | 72 | **15** | 21 | 7 | 2 | 6 |
| MetaCoAG | 17 | 0 | **0** | 34 | **6** | 3 | 69 | 8 | **13** | 7 | 3 | **0** |
| UNITIGBIN | **20** | 0 | 1 | **43** | 4 | 4 | **76** | 11 | 14 | 7 | 5 | 2 |

**Assessing performance using CheckM2.** We utilize CheckM2 (Chklovski et al., 2023) to further assess the performance of UNITIGBIN and the baseline methods. As an illustration, we focus on three simulated datasets assembled by metaSPAdes. Our experimental setup aligns with the guidelines provided in MetaCoAG (Mallawaarachchi & Lin, 2022), where High-quality (HQ) bins are defined by precision > 90 and recall > 80. Medium-quality (MQ) bins exhibit precision > 80 and recall > 50, while the remaining bins are categorized as Low-quality (LQ) bins. The CheckM2 repository can be accessed at `https://github.com/chklovski/CheckM2`, and the execution command is $ `checkm2 predict --threads 64 --input /input_dir/* --output-directory /output_dir/`. The detailed experimental results are compiled in Table A7. UNITIGBIN demonstrates superior performance compared to other baselines, securing a top-tier position. As an instance, UNITIGBIN identifies 20 high-quality bins in the Sim20G dataset, a number equivalent to that achieved by MaxBin2 and VAMB. In the Sim50G dataset, UNITIGBIN secures 41 high-quality bins, slightly surpassing the count achieved by MaxBin2 (40). In the Sim100G dataset, UNITIGBIN acquires 69 high-quality bins, slightly fewer than the count achieved by SemiBin (70). UNITIGBIN continues to show good performance in evaluations using CheckM2.

**Additonal comparison on CAMI dataset.** While our paper extensively employs various simulated and real-world datasets to assess the performance of UNITIGBIN in comparison to state-of-the-art baselines, we also undertake benchmarking on a specific dataset (Urogenital tract, referred toe as CAMI_UG) from the toy Human Microbiome project of the second CAMI challenge (Sczyrba et al., 2017; Meyer et al., 2022). Additionally, we adhere to the experimental setup outlined in MetaCoAG and employ CheckM as the evaluation tool for conducting preliminary comparisons. In this part, our primary choice for a comparative method is VAMB (Nissen et al., 2021), a deep learning-based binning tool. A summary of the experimental results is presented in Table A8. UNITIGBIN slightly surpasses the performance of the VAMB model, yielding 35 high-quality bins and 26 medium-quality bins. UNITIGBIN can bin a greater number of contigs than VAMB, with a runtime of approximately 40 minutes (For detailed information about the running environment, please refer to Section A.3.).

Table A7: Evaluation results by UNITIGBIN and baselines (CheckM2).

| Methods | Sim20G | | | Sim50G | | | Sim100G | | |
|---|---|---|---|---|---|---|---|---|---|
| | HQ↑ | MQ↑ | LQ↓ | HQ↑ | MQ↑ | LQ↓ | HQ↑ | MQ↑ | LQ↓ |
| MetaBAT2 | 4 | **4** | 80 | 7 | 7 | 208 | 4 | 6 | 22 |
| MaxBin2 | **20** | 0 | **1** | 40 | 2 | **3** | 65 | 12 | 18 |
| SemiBin | 19 | 1 | 2 | 39 | 2 | 9 | **70** | 8 | 20 |
| VAMB | **20** | 0 | 2 | 35 | 1 | 10 | 68 | 6 | 13 |
| GraphMB | 10 | 1 | 23 | 14 | 0 | 73 | 22 | 3 | 170 |
| CCVAE | 13 | 1 | 37 | 15 | 0 | 71 | 35 | 1 | 119 |
| RepBin | 18 | 1 | **1** | 17 | **9** | 21 | 25 | **14** | 39 |
| MetaCoAG | 18 | 1 | **1** | 39 | 4 | 4 | 67 | 12 | **4** |
| UNITIGBIN | **20** | 0 | **1** | **41** | 8 | 9 | 69 | 12 | 14 |

To apply UNITIGBIN to the CAMI_UG dataset, we configure the 'nbatch' parameter as 5, indicating the division of the original unitig-level assembly graph into 5 subgraphs. While we have an optimization function for joint-unitigs pairs ($p$-Batch), increasing the number of batches may potentially introduce more errors. Thus, there is still a need for efforts to design more efficient binning models on the unitig-level assembly graphs, especially when dealing with large-scale metagenomic data.

Table A8: Evaluation results by UNITIGBIN and VAMB on the CAMI_UG dataset.

| Methods | Binned Contigs | Detected Bins | CAMI_UG | | |
|---|---|---|---|---|---|
| | | | HQ↑ | MQ↑ | LQ↓ |
| VAMB | 21,398 | 100 | 34 | 10 | 56 |
| UNITIGBIN | 49,879 | 121 | 35 | 26 | 53 |

