# OpenReview forum: "Encoding Unitig-level Assembly Graphs with Heterophilous Constraints for Metagenomic Contigs Binning"
_ICLR.cc/2024/Conference — ICLR 2024 poster_

### Official Review · Reviewer_UkJP · 2023-10-31

**Soundness:** 3 good
**Presentation:** 3 good
**Contribution:** 2 fair
**Rating:** 5
**Confidence:** 3

**Summary:**

* A tool for metagenomic binning (clustering) has been developed, consisting of two main components: a learning stage where representations of unitigs (nodes) and contigs (paths) from the unitig assembly graph are considered, along with the constraints that prevent two contigs from belonging to the same bin; and a binning stage where the labels are initialized, propagated, and refined.
* The primary novelty of the proposed method arises from the direct learning approach applied to the unitig assembly graph, rather than reconstructing a contig assembly and learning from it. Additionally, a clustering method has been introduced as a replacement for Kmeans, leveraging the learned representations to address the issue of imbalanced bin size.
* UnitigBin has been evaluated on 12 datasets against baseline methods. The binning results have been assessed by evaluating both the completeness and contamination of the bins using checkM and by evaluating the accuracy of clustering via Precision, Recall, F1, and ARF scores.

**Strengths:**

* The evaluation is comprehensive. It compares different categories of baseline binning methods, including traditional approaches, deep-learning-based binning tools, and assembly graph-based binning tools. Ablation studies and runtime analyses were also conducted, demonstrating the usefulness and efficiency of each component of the proposed framework.
* The binning results from the proposed method show a significant improvement over the baselines on both measures, the completeness and contamination of the bins using checkM and the accuracy of clustering via Precision, Recall, F1, and ARF scores.
* A p-batch training technique is proposed to address the computational challenge without compromising efficiency.

**Weaknesses:**

* It shows that the overall performance is heavily reliant on the choice of downstream clustering methods. Using Kmeans on the learned representation yields surprisingly poor results, which raises the question of whether the representation learning is beneficial in this application.
* Considering the additional computational complexity introduced when learning directly from Unitig assembly graphs, it is not entirely clear why the proposed method can capture additional information that may have been overlooked in prior work that learned from reconstructed contig assembly graphs.

**Questions:**

* Is there any evidence or an intuitive explanation of what additional information is captured by learning directly from the unitig assembly graph instead of the reconstructed contig assembly graph, and how this contributes to the improvement in binning performance? Conducting an ablation study by running a similar framework directly on the unitig assembly graph and demonstrating performance differences could be helpful for illustration.

---

> ### Author Response · Authors · 2023-11-22
>
> Thanks for the comments.
>
> Q1: Applying K-means to the learned embeddings often yields poor performance, primarily due to two reasons: a) imbalanced bin sizes (i.e., varying from a couple to hundreds), and b) inefficient incorporation of information from constraints derived from single-copy marker genes. To tackle these challenges, we designed a combined binning operation that integrates contig-level label propagation while maximizing constraint satisfaction (refer to equation 5).
> It is a good question to ponder whether representation learning is beneficial for metagenomic contig binning. From my perspective, the primary strength of utilizing representation learning techniques, as opposed to traditional algorithms, lies in the efficient embedding of all relevant information together, bypassing the need for intricate algorithm design and moving towards large language models. Within our model, it is possible to concurrently represent the unitig-level assembly graph, nucleotide sequences, and supplementary information from single-copy marker genes.
> In Figure 6, our ablation study on Sim100G reveals that UnitigBin-Learning (applying the K-Means algorithm to learning representations) yields unsatisfactory results. This is primarily attributed to the large number of bins and the challenge of imbalanced bin sizes, ranging from 3 to 652. We also assessed UnitigBin-Learning on Sim20G, a dataset with fewer bins and a less pronounced issue of imbalanced bin sizes. We achieved an F1 score of 0.93 (second only to the 0.97 obtained by RepBin) and an ARI of 0.87 (comparable to 0.96 for VAMB and 0.95 for RepBin). These results demonstrate competitiveness, significantly surpassing non-representation learning-based algorithms like MetaCoAG, which achieved F1 and ARI values of 0.54 and 0.79, respectively. Therefore, models capable of producing effective representations prove advantageous for metagenomic contig binning.
>
> Q2: We choose to learn the unitig-level assembly graph rather than the contig-level assembly graph to preserve the original data and avoid introducing errors in constructing the contig-level assembly graph from the unitig-level assembly graph.  The output from assemblers (such as metaSPAdes and metaFlye) is essentially a unitig-level assembly graph. Previous graph-based models devised strategies to construct a contig-level assembly graph from the unitig-level assembly graph to simplify it for the task of binning contigs. However, this construction process introduces errors, indicating that linked edges in the contig-level assembly graph may not necessarily belong to the same bin (e.g., the error rate for Sim50G is 7.3%). The motivation behind using the assembly graph is rooted in the homophily assumption, where linked unitigs/contigs are likely to belong to the same bin. The introduction of erroneous edges in the contig-level assembly graph adds complexity to embedding or learning. To circumvent these errors, we directly focus on the original unitig-level assembly graph.

---

### Official Review · Reviewer_WrPh · 2023-11-01

**Soundness:** 2 fair
**Presentation:** 2 fair
**Contribution:** 2 fair
**Rating:** 3
**Confidence:** 4

**Summary:**

In this paper, the authors introduce a novel binning tool, UNITIGBIN, which encodes unitig-level assembly graphs while considering constraints generated by single-copy marker gene information. After obtaining the embeddings of the sequences, UNITIGBIN bins the contigs (formed by combining one or multiple connected unitigs) using a binning framework consisting of 1) binning initialization, 2) iterative matching, 3) propagating and 4) refining. The results from datasets with assembly graphs generated by two different assemblers demonstrate that UNITIGBIN outperforms the compared methods in overall.

**Strengths:**

This paper offers a practical approach for transitioning from contig-level representation learning to unitig-level.

**Weaknesses:**

The novelty appears somewhat limited, as the approach primarily combines existing representation learning techniques and the constraint-based graph representation learning-based binning method RepBin, albeit with the conversion of assembly graphs from contig-level to unitig-level. Additionally, the binning steps appear to be a combination of Metacoag (binning initialization) and RepBin (propagating and refining) with limited improments. The authors should provide a more detailed and distinct description of the similarities and differences.

The experiments are inadequate in terms of comparing the results obtained from unitig-level assembly graphs with those from contig-level assembly graphs. These experiments are significant for demonstrating the transition from unitig-level assembly graphs to contig-level assembly graphs.

The experiments are insufficient. Most of the datasets are small-scale single-sample datasets. For a more robust comparison of binning performance, it is recommended to utilize the CAMI I and CAMI II benchmark datasets.  In addition, the recently introduced Metadecoder has shown its effectiveness as a method. SemiBin2, an improvement over Semibin1, presents a novel approach specifically designed for long read data. It is highly recommended that the authors include these methods in the benchmark analysis.

We recommend that the authors employ Checkm2 to assess MAG quality, as it outperforms the original version of CheckM in predicting MAG quality, even for those with sparse genomic representation or reduced genome size.

This method is tailored to address a particular problem and may pose challenges when attempting to adapt it for use in different domains.

**Questions:**

Please provide more details about the datasets, for example, the number of the sequencing samples, the way to obtain the ground truth.
The definition of single-copy marker genes and the introduction of semibin appear to lack precision

---

> ### Author Response · Authors · 2023-11-22
>
> Thanks for the comments.
>
> Q1: UnitigBin cannot be viewed simply as an amalgamation of existing models. UnitigBin-Learning differs significantly from RepBin-Learning in that it not only provides them with different inputs. Furthermore, the initialization, propagation, and refinement processes within the Binning phase clearly differentiate UnitigBin from MetaCoAG and RepBin. We will highlight the contributions of our proposed UnitigBin model.
>
> a) Modeling unitig-level assembly graph: We chose to learn unit-level assemblies rather than contig-level assemblies to preserve the original data and avoid introducing errors when constructing the contig-level assembly graph from the unitig-level assembly graph. The output of assemblers such as metaSPAdes and metaFlye is essentially a unitig-level assembly graph. Previous graph-based models devised strategies to construct contig-level assembly graphs from unit-level assembly graphs to simplify the contig binning task. However, this construction process introduces errors indicating that linked edges in the contig-level assembly graph may not necessarily belong to the same bin. The motivation behind using assembly diagrams is rooted in the assumption of homogeneity, where linked units/contigs may belong to the same bin. Introducing false edges into the contig-level assembly graph increases the complexity of embedding or learning. To avoid these errors, we focus directly on the original unitig-level assembly graph.
>
> b）Learning: In the unitig-level assembly graph, nodes correspond to unitigs, and edges represent overlapping information between unitigs. Contigs are represented as paths in the unit-level assembly graph. In the assembly graph, nodes are contigs, and edges represent the overlapping information between contigs. 1) UnitigBin adopts a generative encoder-decoder framework as its main architecture, avoiding the need for negative graph sampling (time and memory consumption in large-scale assembly of graphs). RepBin, on the other hand, utilizes a contrastive graph learning framework. 2) UnitigBin incorporates heterophilious constraints by introducing a triplet Gaussian constraints optimization at the unitig level, while RepBin calculates the Euclidean distance between pairs of nodes in the contig level.
>
> c) Binning: a) UnitigBin employs a simple yet efficient matching algorithm to initialize constrained contigs by calculating the embedding and composition. In contrast, MetaCoAG initializes binnings by constructing a Weighted Bipartite Graph, finding a Minimum-Weight Full Matching, and dynamically adjusting bins based on three cases. b) UnitigBin performs label propagation at the unitig level, while taking into account heterogeneous constraints by designing a loss function that satisfies the constraints. RepBin, on the other hand, performs label propagation at the contig level and does not consider information from constraints. c) UnitigBin also uses a refinement algorithm to adjust the final binning results to minimize the number of constraint violations. In contrast, RepBin does not include any refining steps.
>
> Q2: Thank you for the recommendation. We will include Metadecoder as the baseline and apply SemiBin2 to long-read datasets. Unfortunately, due to time constraints, we are unable to execute these two baselines. Due to Metadecoder's inferior performance compared to typical methods such as MetaBAT2 and SemiBin in several studies, and considering it is not a graph-based method, our paper primarily concentrates on graph-based methods and representative binning tools.
>
> Q3: Thanks for the suggestion. We'll incorporate CheckM2 as an additional evaluation tool for long-read datasets.
>
> Q4: Our proposed learning and clustering framework may also hold potential for exploration in various other microbiome-related domains, such as microbial sequences recovery. In mixed microbial communities, a diverse array of bacterial chromosomes and mobile genetic elements coexists, including microeukaryotes, plasmids, bacteriophages, and more. Accurately identifying these sequencing reads/contigs and reconstructing species is crucial not only for unraveling the functions of these microbiomes but also for monitoring microbial changes in oil and other contexts. It is feasible to integrate a graph representation learning framework into the encoding of the assembly graph, framing microbial sequence identification as a graph learning and identification problem.
>
> Q5: Thank you for the suggestion. We will incorporate more details about the dataset.

---

### Official Review · Reviewer_22Pk · 2023-11-03

**Soundness:** 3 good
**Presentation:** 3 good
**Contribution:** 2 fair
**Rating:** 5
**Confidence:** 2

**Summary:**

The paper presented a graph neural net based approach for the application of binning metagnomic contigs, ie. clustering genomic subsequences into bins representing a constituent organism. The paper showed numerical results where the presented approach outperformed the other tested approaches on 12 public datasets.

**Strengths:**

The application seems to be novel by introducing the popular-studied approach of graph neural nets. The experiment includes 12 datasets and compares against 3 categories of tools.

**Weaknesses:**

I like how the authors keep the section of Problem Statement, but I don’t follow the exact problem to be solved however. The paragraph on Page 3 in Problem Statement only talked about the notation but does not specify what the task is, or what the desired output should be. Further, it wasn’t clear what objective is aimed at – which also motivates my question for how we make sure the objective function in each component maps to the objective of binning?

Separately, in the experiment setup, it’s unclear how the training/testing/validation is conducted; also while the results reported in Table 2 are promising, it’s unclear how significant they are.

Since one motivation described in the Intro is that the presented approaches will respect the constraints – it’d be good to explicitly evaluate how it was done as compared to other methods, either in an explicit metric (or perhaps I mis-interpret the existing one?) or through an example.

**Questions:**

1) Suggest to motivate the readers more by explicitly describing the problem formulation including what objective is desired; ideally this can be formulated as an mathematical objective function and tied to the choice of loss function in the later learning component.

2) Suggest to describe the experiment setup including what training/testing/validation data is included and clarify the statistical significance of the result comparison; if this is not feasible, provide justification and identify future steps.

3) It’s not clear from the experiment results why particularly GNN is needed – suggest to provide perhaps case studies to show how GNN is able to capture the information that the other approaches cannot, since introducing GNN is considered as the main contribution of the work.

---

> ### Author Response · Authors · 2023-11-22
>
> Thanks for the comments.
>
> Q1: The goal of binning metagenomic contigs is to group nucleotide sequences (e.g., ATCG) into separate clusters. Our input data includes the assembly graph at the unitig level, nucleotide sequences, and additional constraints obtained from single-copy marker genes.
> We simplify the clustering challenge of sequences and frame it as a task of graph learning and constraint satisfaction. In detail, we aim to incorporate two key pieces of information: a)homophily information within the assembly graph, indicating that unitigs/contigs linked together in the assembly graph are more likely to belong to the same bin, and b)heterophilious relations of constraints, stating that unitigs/contigs within the same constraint set must belong to distinct bins. The ideal situation for contig binning is that all contigs are grouped into distinct clusters corresponding to the constituent genomes and satisfy previous heterophilic constraints.
> i) In the learning phase, our primary focus is on acquiring the homophily structural information of the unitig-level assembly graph while simultaneously integrating heterophilious constraints in a soft manner, accomplished through the inclusion of a penalty function (refer to the equation 3). The objective function of the learning phase mainly consists of two components: the loss of the reconstructed graph that captures homogeneous information in the assembled graph and the loss of optimizing heterogeneous constraints to maximize constraint satisfaction.
> ii) During the binning phase, our attention shifts to optimizing the heterophilous constraints and synergizing label propagation with maximizing constraint satisfaction (refer to equation 5). The primary components of the binning phase's objective function include labeling unannotated contigs using the homogeneous assembly graph and incorporating a loss term to maximize constraint satisfaction. The binning process, specifically the minimization of constraint violations, is illustrated in Figure A3(b). It can be observed that as the training/binning epochs increase, the number of violating heterophilious constraints decreases.
> iii) Finally, as there are still some binning results violating the constraints, we introduce a refinement method to eliminate such violations.
>
> Q2:The binning of metagenomic contigs emphasize the extraction of features from the nucleotide sequences and assembly graph, possibly incorporating additional information from references or single-copy marker genes. And then organize these contigs into distinct clusters corresponding to the constituent genomes. Thus, contigs binning can be conceptualized as an unsupervised clustering problem.
> In our paper, we utilize 12 datasets—six assembled by metaSPAdes and six assembled by metaFlye. We execute our proposed UnitigBin along with selected baselines on these datasets to generate the final binning results. In the experiment, we primarily employ CheckM v1.1.3 and AMBER v2.0.2 to assess the performance of UnitigBin and the baseline methods. In Table 2, all runs of UnitigBin and the baseline methods are performed five times on Sim20G, and the best binning assignment (evaluated using the AMBER tool) is reported. Additionally, we also present the average values along with the corresponding standard deviation values. Significance tests are also conducted, and the results are presented below.
> | Methods   | F1(bp)         | F1(seq)        | ARI(bp)        | ARI(seq)       | C<5;P>90  |
> | --------- | -------------- | -------------- | -------------- | -------------- | --------- |
> | Maxbin2   | 0.985(0.005)   | 0.634(0.010)   | 0.987(0.003)   | 0.784(0.008)   | 19.3(0.6) |
> | SemiBin   | 0.982(0.007)   | 0.541(0.004)   | 0.954(0.024)   | 0.410(0.008)   | 19.0(0.0) |
> | VAMB      | 0.983(0.007)   | 0.604(0.012)   | 0.985(0.009)   | 0.969(0.007)   | 18.3(0.6) |
> | MetaBAT2  | 0.629(0.025)   | 0.313(0.011)   | 0.400(0.009)   | 0.224(0.008)   | 4.0(0.0)  |
> | GraphMB   | 0.950(0.031)   | 0.603(0.022)   | 0.552(0.012)   | 0.364(0.017)   | 9.3(0.6)  |
> | CCVAE     | 0.981(0.004)   | 0.617(0.004)   | 0.790(0.004)   | 0.431(0.034)   | 13.3(0.6) |
> | RepBin    | 0.955(0.012)   | 0.447(0.011)   | 0.959(0.004)   | 0.148(0.009)   | 15.7(0.6) |
> | MetaCOAG  | 0.955(0.006)   | 0.591(0.002)   | 0.988(0.003)   | 0.783(0.004)   | 14.7(0.6) |
> | UnitigBin | 0.995(0.005)\* | 0.953(0.003)\* | 0.992(0.003)\* | 0.780(0.006)\* | 19.7(0.6) |

---

> ### Author Response · Authors · 2023-11-22
>
> Q3: One of the objectives of our proposed UnitigBin model is to maximize the satisfaction of heterophilious constraints. Therefore, we calculate the ratio of binning results that violate constraints. The process of minimizing the number of violating constraints is illustrated in Figure A3(b). Additionally, the ratio of violating constraints achieved by UnitigBin and selected baselines is presented in the table below (on Sim100G). UnitigBin fulfills nearly all constraints, with only a 0.9% violation of heterophilious constraints, a lower rate compared to other baseline methods.
> | MaxBin2     | SemiBin     | VAMB        | MetaBAT2 | GraphMB      | CCVAE        | RepBin      | MetaCoAG    | UnitigBin   |
> | ----------- | ----------- | ----------- | -------- | ------------ | ------------ | ----------- | ----------- | ----------- |
> | 1712/127449 | 2226/180044 | 2623/152948 | 37/1313  | 42244/170476 | 22627/167211 | 8854/177910 | 2069/153026 | 1686/182366 |
> | 1.34%       | 1.24%       | 1.92%       | 2.82%    | 24.80%       | 13.53%       | 4.98%       | 1.35%       | 0.90%       |
>
> Q4: Traditional contig binning methods depend on statistical information derived solely from contigs, neglecting homophily information present in the assembly graph. Homophily suggests that sequences connected in the assembly graph are more likely to belong to the same species. Typically, statistics-based approaches tend to exclude short sequences, often shorter than 1,000 bp, due to the instability of features extracted from nucleotide sequences. GNNs can leverage the homophily present in the assembly graph to assign labels to these short sequences.

---

### Meta-Review · Area_Chair_CkgL · 2023-12-05

**Metareview:**

The paper proposes a novel binning tool named UnitigBin, which leverages representation learning on unitig-level assembly graphs while adhering to heterophilious constraints imposed by single-copy marker genes, ensuring that constrained contigs cannot be grouped together. The application of this method is to more accurately parse short reads and assign them to the organisms that gave rise to them.

The paper condenses a large number of methodological details and experiments into a relatively short format. The combination of graph learning with constraints in a parallelizable algorithm seems to be a novel contribution that is well-motivated by the application domain.

One reviewer raised concerns about the scale and applicability of the experiments. The concern was acknowledged by the authors and they promised to address it in the final version. There was a concern that the novelty was limited as it is combines "existing representation learning techniques and the constraint-based graph representation learning-based binning method RepBin". The combination of existing techniques in ways that are adapted to and relevant for the application domain is perhaps a strength and not necessarily a weakness unless the combination is obvious. For this methodology to gain traction in the domain, software implementing the strategy would need to be made available. In appendix A3 the authors present their implementation; the pipeline could be containerized for easier reproducibility.

**Justification For Why Not Higher Score:**

The reviewers noted some weaknesses that are well-justified in their reviews and summarized in my meta-review.

**Justification For Why Not Lower Score:**

The work seems to be well-motivated and in the extensive experiments performs well on real and simulated data sets. Weaknesses related to clarity of the problem statement - the problem being approached is a common one in the domain, though possibly less familiar to other audiences. The authors have addressed concerns about what the effect of not violating constraints has on the performance. The authors responded to the reviewers' concerns regarding experiments and promised to address them in the final version.

---

### Decision · Program_Chairs · 2024-01-16

Accept (poster)